# Efficient characterizations of multiphoton states with an ultra-thin optical device

Kui An [1,8], Zilei Liu [2,3,8], Ting Zhang[1], Siqi Li[2], You Zhou[4,5], Xiao Yuan [6], Leiran Wang [2,3], Wenfu Zhang [2,3], Guoxi Wang [2,3] ✉ & He Lu [1,7] ✉

Metasurface enables the generation and manipulation of multiphoton entanglement with flat optics, providing a more efficient platform for large-scale photonic quantum information processing. Here, we show that a single metasurface optical device would allow more efficient characterizations of multiphoton entangled states, such as shadow tomography, which generally requires fast and complicated control of optical setups to perform information-complete measurements, a demanding task using conventional optics. The compact and stable device here allows implementations of general positive operator valued measures with a reduced sample complexity and significantly alleviates the experimental complexity to implement shadow tomography. Integrating self-learning and calibration algorithms, we observe notable advantages in the reconstruction of multiphoton entanglement, including using fewer measurements, having higher accuracy, and being robust against experimental imperfections. Our work unveils the feasibility of metasurface as a favorable integrated optical device for efficient characterization of multiphoton entanglement, and sheds light on scalable photonic quantum technologies with ultra-thin optical devices.

Metasurface, an ultra-thin and highly integrated optical device, is capable of full light control and thus provides novel applications in quantum photonics[1]. In photonic quantum information processing, multiphoton entanglement is the building block for wide range of tasks, such as quantum computation[2], quantum error correction[3], quantum secret sharing[4,5], and quantum sensing[6]. Recent investigations highlighted the feasibility of metasurface in generation[7,8], manipulation[9–11], and detection[12,13] of multiphoton entanglement, indicating metasurface as a promising technology of ultra-thin optical device for large-scale quantum information processing.

Characterization of multiphoton entanglement provides diagnostic information on experimental imperfections and benchmarks our technological progress towards the reliable control of large-scale photons. The standard quantum tomography (SQT)[14] requires an exponential overhead with respect to the system size. Recently, more efficient protocols have been proposed and demonstrated with fewer measurements, such as compressed sensing[15,16], adaptive tomography[17–19] and self-guided quantum tomography (SGQT)[20–22]. Shadow tomography, which was first proposed by Aaronson et al.[23] and then concreted by Huang et al.[24], efficiently predicts functions of the quantum states instead of state reconstruction. Huang's protocol[24] is hereafter referred as shadow tomography. Shadow tomography is efficient in estimation of quantities in terms of observable (polynomial), including nonlinear observables such as purity and Rényi entropy[25–28], which is of particular interest in detecting multipartite

[1]School of Physics, State Key Laboratory of Crystal Materials, Shandong University, Jinan 250100, China. [2]State Key Laboratory of Transient Optics and Photonics, Xi'an Institute of Optics and Precision Mechanics, Chinese Academy of Sciences, Xi'an 710119, China. [3]University of Chinese Academy of Sciences, Beijing 100049, China. [4]Key Laboratory for Information Science of Electromagnetic Waves (Ministry of Education), Fudan University, Shanghai 200433, China. [5]Hefei National Laboratory, Hefei 230088, China. [6]Center on Frontiers of Computing Studies, Peking University, Beijing 100871, China. [7]Shenzhen Research Institute of Shandong University, Shenzhen 518057, China. [8]These authors contributed equally: Kui An, Zilei Liu. ✉e-mail: wangguoxi@opt.ac.cn; luhe@sdu.edu.cn

entanglement[29–32] and thus is helpful in benchmarking the technologies towards generation of genuine multipartite entanglement[33–35]. Nevertheless, shadow tomography generally requires the experimental capability of performing information-complete measurements, leading to the consequence that the time of switching experimental setting is much longer than that of data acquisition. A potential solution is to replace the unitary operations and projective measurements with positive operator valued measures (POVMs), which is capable to extract complete information in a single experimental setting[36–38]. The POVM significantly alleviates the experimental complexity to perform shadow tomography, and thus enables the real-time shadow tomography, i.e., an experimentalist is free to stop shadow tomography at any time. However, a compact and scalable implementation of POVM in optical system is still technically challenging. On the other hand, shadow tomography is not able to easily predict the properties that cannot be directly expressed in terms of observables (polynomial) such as von Neumann entropy $S(\rho) = -\mathrm{Tr}(\rho\log\rho)$, which is key ingredient in topological entanglement entropy[39,40].

In this work, we report an implementation of POVM enabled by a metasurface, which is based on planar arrays of nanopillars and able to provide complete control of polarization. The POVM we achieved allows to implement real-time shadow tomography, and observe the shadow norm that determines sample complexity. Moreover, we show that the metasurface-enabled shadow tomography can be readily equipped with other algorithms, enabling the unexplored advantages of shadow tomography. In particular, we propose and implement shadow tomography optimized by simultaneous perturbation stochastic approximation (SPSA)[41], the so-called self-learning shadow tomography (SLST). SLST efficiently returns a physical state with high accuracy against the metasurface-induced imperfections, which can be further used to calculate the quantities that cannot be expressed in terms of directly observable. We also implement robust shadow tomography[42] to show the robustness of reconstruction against the engineered optical loss.

## Results

### Shadow tomography with POVM

We start by briefly reviewing the shadow tomography with POVM. Considering a 2-level (qubit) quantum system, a set of $L$ rank-one projectors $\{|\psi_l\rangle\langle\psi_l| \in \mathbb{H}_2\}_{l=1}^{L}$ is called a quantum 2-design if the average value of the second-moment operator $(|\psi_l\rangle\langle\psi_l|)^{\otimes 2}$ over the set is proportional to the projector onto the totally symmetric subspace of two copies[43]. Each quantum 2-design is proportional to a POVM $\mathbf{E} = \{\frac{2}{L}|\psi_l\rangle\langle\psi_l|\}_{l=1}^{L}$ with the element $E_l = \frac{2}{L}|\psi_l\rangle\langle\psi_l|$ being positive semi-definite and satisfying $\sum_{l=1}^{L} E_l = \mathbb{1}_2$. Note that quantum 1-design is sufficient to form a POVM but is not always information-complete for tomography, such as the measurement on computational basis $\{|0\rangle, |1\rangle\}$. Measuring a quantum state $\rho$ using POVM $\mathbf{E}$ results one $l \in [L]$ outcome with probability $Pr(l|\rho) = \mathrm{Tr}(E_l\rho)$ according to Born's rule. The POVM $\mathbf{E}$ together with the preparation of the corresponding state $|\psi_l\rangle$ can be viewed as a linear map $\mathcal{M}: \mathbb{H}_2 \to \mathbb{H}_2$, and the 'classical shadow' is the solution of least-square estimator with single experimental run,

$$\hat{\rho}_l^{(m)} = \mathcal{M}^{-1}(|\psi_l\rangle\langle\psi_l|) = 3|\psi_l\rangle\langle\psi_l| - \mathbb{1}_2. \qquad (1)$$

For an $N$-qubit state, the classical shadow is the tensor product of simultaneous single-qubit estimations $\hat{\rho}^{(m)} = \bigotimes_{n=1}^{N}\hat{\rho}_{l_n}^{(m)}$ with $l_n$ being the outcome of $n$-th qubit, and one has $\mathbb{E}[\hat{\rho}^{(m)}] = \rho$. Repeating the POVM $M$ times (experimental runs), one has a collection of classical shadows $\{\hat{\rho}^{(m)}\}_{m=1}^{M}$, which is further inquired for estimation of various properties of the underlying state. See Supplementary Note 1A for more details.

### Implementation of POVM with metasurface

In our experiment, we focus on the POVM on polarization-encoded qubit, i.e., $|0(1)\rangle = |H(V)\rangle$ with $|H(V)\rangle$ being the horizontal (vertical) polarization, and consider POVM of $L = 6$ and $|\psi_l\rangle \in \{|H\rangle, |V\rangle, |+\rangle, |-\rangle, |R\rangle, |L\rangle\}$ with $|\pm\rangle = (|H\rangle \pm |V\rangle)/\sqrt{2}$ and $|R(L)\rangle = (|H\rangle \pm i|V\rangle)/\sqrt{2}$. The corresponding POVM $\mathbf{E}_{octa}$ is described by a symmetric polytope of an octahedron on Bloch sphere as shown in Fig. 1a. To realize $\mathbf{E}_{octa}$, we

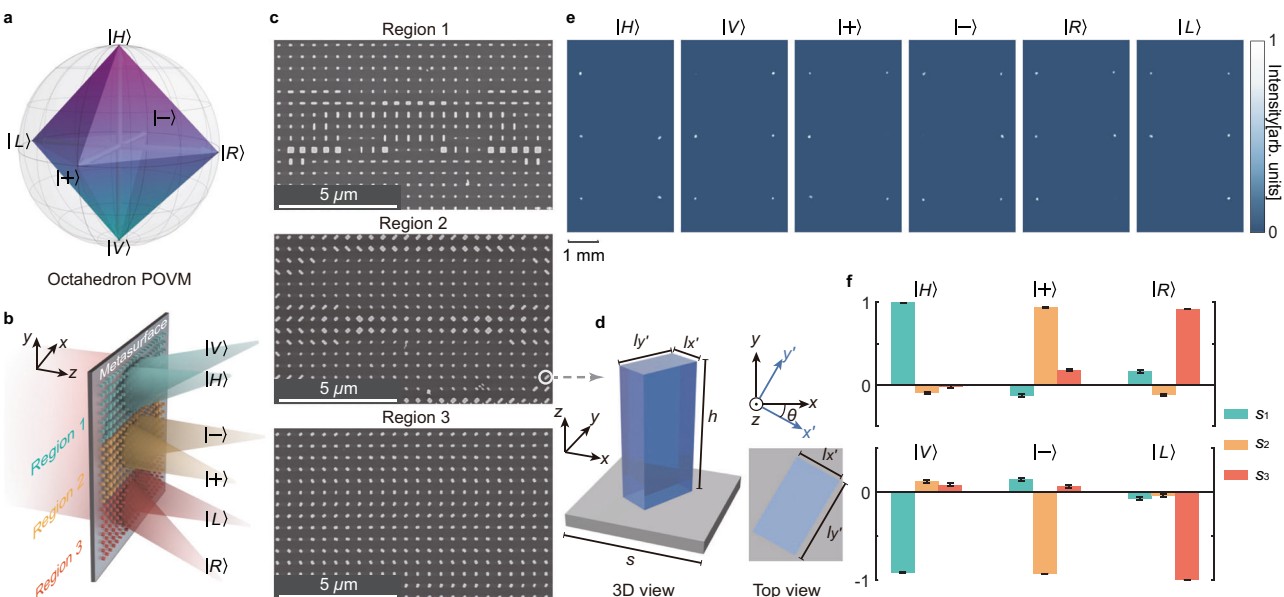

**Fig. 1 | The metasurface-enabled octahedron positive operator valued measure (POVM) $\mathbf{E}_{octa}$. a** The elements in $\mathbf{E}_{octa}$ are projectors on states $|H\rangle$, $|V\rangle$, $|+\rangle$, $|-\rangle$, $|R\rangle$ and $|L\rangle$ respectively, which form a symmetric polytope of an octahedron on Bloch sphere. **b** The metasurface to realize $\mathbf{E}_{octa}$, green, yellow and red blocks on the metasurface represent nanopillars with different arrangements. **c** The scanning electron microscopy images of the fabricated nanopillars in three regions. **d** Schematic drawing of single nanopillar that is fabricated with same height of 700 nm but different $(\theta, l_{x'}, l_{y'})$. **e** The measured distribution of intensity on focal plane with input polarization of $|H\rangle$, $|V\rangle$, $|+\rangle$, $|-\rangle$, $|R\rangle$ and $|L\rangle$, respectively. **f** The reconstructed Stokes parameters $(s_1, s_2, s_3)$ from data collected in (**e**) and the error bars indicate standard deviations of reconstructed Stokes parameters.

design and fabricate a 210 $\mu m \times$ 210 $\mu m$ polarization-dependent metasurface that splits incident light into six directions corresponding to projection on $|H\rangle$, $|V\rangle$, $|+\rangle$, $|-\rangle$, $|R\rangle$ and $|L\rangle$ with equal probability (shown in Fig. 1b). Note that projection on $|\psi_l\rangle$ with equal probability is guaranteed with post-selection to eliminate the mode mismatch between incident light (Gaussian beam) and metasurface (square) (see Supplementary Note 5 for details). The metasurface is an array (with square pixel of $s = 500$ nm) of single-layer amorphous silicon nanopillars on quartz substrate as shown in Fig. 1c, d. The nanopillars are with the same height of 700 nm but different $l_{x'}$, $l_{y'}$ and orientation $\theta$ relative to the reference coordinate system. In this sense, a single nanopillar can be regarded as a waveguide with different rectangular cross profile that exhibits corresponding effective birefringence, leading to spatial separation between orthogonal polarizations[44]. The metasurface is divided into three regions with same size of 210 $\mu m \times$ 70 $\mu m$ but different arrangement of nanopillars, i.e., $(\theta, l_{x'}, l_{y'})$. By carefully designing the arrangement of nanopillars, we can realize spatial separation of $|H\rangle/|V\rangle$, $|+\rangle/|-\rangle$ and $|R\rangle/|L\rangle$, respectively. To validate the capability of fabricated metasurface to perform information-complete measurement, we test metasurface with input states of $|\psi_l\rangle$ and measure the distribution of output intensity on focal plane. The results are shown in Fig. 1e, according to which we reconstruct the Stokes parameters $(s_1, s_2, s_3)$ shown in Fig. 1f. Compared to the ideal values, the average errors of reconstructed Stokes parameters are $0.101 \pm 0.005$, $0.086 \pm 0.005$ and $0.073 \pm 0.005$, respectively. These errors are mainly caused by the discretization of phase front in design, which inevitably introduces higher-order deflections[45] (see Supplementary Note 4 for details of metasurface).

## Estimation of observables

We first perform shadow tomography with the fabricated metasurface on single-photon pure state $|\psi_{\gamma,\phi}\rangle = \cos\gamma|H\rangle + \sin\gamma e^{i\phi}|V\rangle$ with $\gamma = 0.91$ and $\phi = 0.12$. As shown in Fig. 2a, the polarization-entangled photons (central wavelength of 810 nm) are generated from a periodically poled potassium titanyl phosphate (PPKTP) crystal placed in a Sagnac interferometer via spontaneous parametric down conversion (SPDC), which is pumped by a laser diode (central wavelength of 405

nm). The generated entangled photons are with ideal form of $|\psi\rangle_\eta = \sqrt{\eta}|HV\rangle + \sqrt{1-\eta}|VH\rangle$, where $\eta$ is determined by polarization of pump light. Projecting one photon of $|\psi\rangle_\eta$ on $|H\rangle$ heralds the other photon on state $|V\rangle$, which can further be transformed to arbitrary $|\psi_{\gamma,\phi}\rangle = \cos\gamma|H\rangle + \sin\gamma e^{i\phi}|V\rangle$ by a combination of electrically-rotated half-wave plate (E-HWP) and quarter-wave plate (E-QWP). Then, the heralded photon passes through the metasurface, and is coupled to six multimode fibers at outputs using an objective lens (OL), a tube lens, and three prisms, respectively. With the collection of classical shadows $\{\hat{\rho}^{(m)}\}_{m=1}^M$, we focus on the estimation of observables in set of 128 single-qubit projections, i.e., $O = |\psi_{\kappa,\nu}\rangle\langle\psi_{\kappa,\nu}| \in \mathbf{O}$ with $|\psi_{\kappa,\nu}\rangle = \cos\kappa|H\rangle + \sin\kappa e^{i\nu}|V\rangle$ being uniformly distributed on Bloch sphere. The estimation of expected value of observable is $\hat{O} = 1/M\sum_{m=1}^M \hat{o}^{(m)}$, where $\hat{o}^{(m)} = Tr(O\hat{\rho}^{(m)})$ is the i.i.d single-shot estimator. Note that $\hat{O}$ converges to the exact expectation value $Tr(\rho O)$ as $M \to \infty$. The error of estimation with metasurface-enabled POVM is indicated by the distance between $\hat{O}$ and ideal expectation $\langle O\rangle = \langle\psi_{\gamma,\phi}|O|\psi_{\gamma,\phi}\rangle$. As shown in Fig. 2b, the maximal distance $\max_{\mathbf{O}} \|\hat{O} - \langle O\rangle\|$ converges to 0.07 with the increase of $M$, which is consistent with the error we obtained in reconstruction of Stokes parameters. In Fig. 2c, we show the real-time estimation of $\hat{O}$ by randomly selecting five $O \in \mathbf{O}$, in which we observe the convergence of $\hat{O}$ after a few hundreds of milliseconds.

The sample complexity of estimation is further characterized by the variance $\text{Var}(\hat{O}) = \text{Var}(\hat{o}) \leq \|O\|_{\text{shd}}^2$. Here the shadow norm $\|O\|_{\text{shd}}^{24}$ is the maximization of $\text{Var}(\hat{o})$ over all possible states $\rho$ to remove the state-dependence. For ideal $\mathbf{E}_{\text{octa}}$, the shadow norm $\|O\|_{\text{shd}}^2 = 0.75$ regardless of the explicit form of $O \in \mathbf{O}$ (see Supplementary Note 1D for deviation of $\|O\|_{\text{shd}}^2 = 0.75$). Experimentally, the variance of a single-shot estimation is

$$\text{Var}(\hat{o}) = \frac{1}{M}\sum_{m=1}^M \left(\hat{o}^{(m)} - \hat{O}\right)^2. \qquad (2)$$

It is impossible to maximize $\text{Var}(\hat{o})$ over all possible $|\psi_{\gamma,\phi}\rangle$ in experiment, so that we prepare totally 20 $|\psi_{\gamma,\phi}\rangle$ that are uniformly distributed on Bloch sphere, forming a state set of $\mathbf{P}$. For each prepared

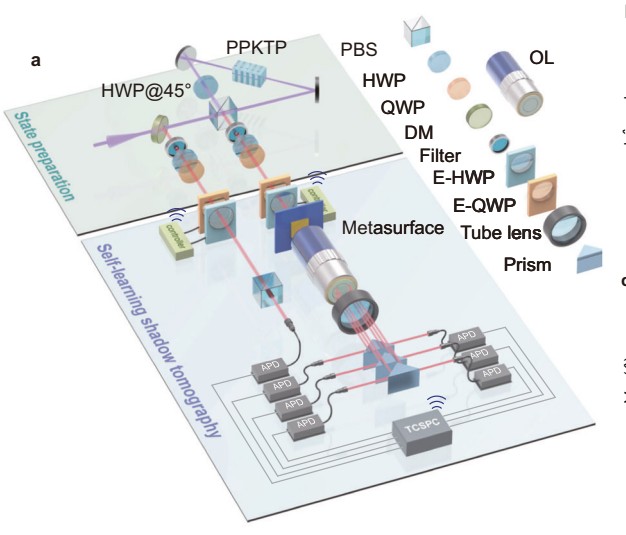

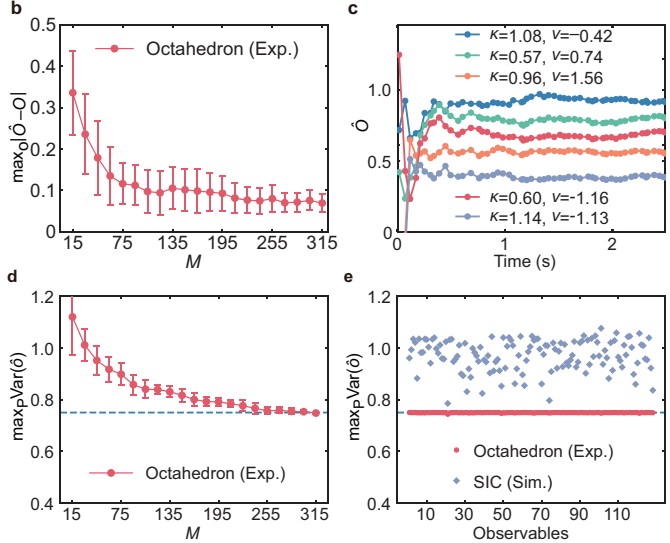

**Fig. 2 | The experimental setup and results of shadow tomography with metasurface-enabled positive operator valued measure (POVM). a** Setup to generate entangled photons and demonstrate shadow tomography with metasurface. PBS: polarizing beam splitter. DM: dichroic mirror. HWP: half-wave plate. QWP: quarter-wave plate. E-HWP: electrically-rotated HWP. E-QWP: electrically-rotated QWP. OL: objective lens. **b** The maximal error in estimation of expectation of $O \in \mathbf{O}$. **c** The real-time estimation of expectation of five randomly selected $O \in \mathbf{O}$. **d** The

results of shadow norm $\max_{\mathbf{P}} \text{Var}(\hat{o})$ for $O = |+\rangle\langle+|$ with different experimental runs. **e** The results of shadow norm for 128 $O \in \mathbf{O}$ (red dots), and the simulated results of shadow norm with symmetric informationally complete (SIC) POVM (blue diamonds). The 128 observables $O \in \mathbf{O}$ are selected according to Haar random. The dots and bars in (**b**) and (**d**) are the mean value and corresponding standard deviations obtained by repeating the experiment 5 times. The abbreviations of Exp. and Sim. indicate experimental results and simulation results respectively.

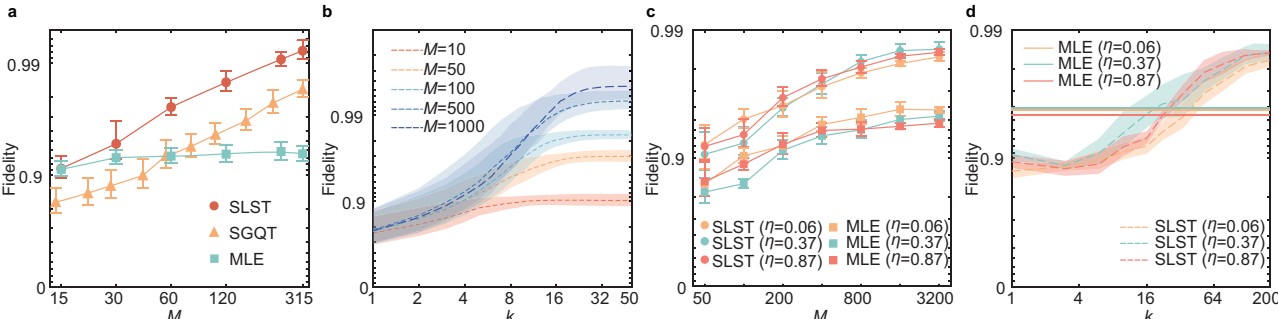

**Fig. 3 | Experimental results of self-learning shadow tomography (SLST) on one-photon and two-photon states. a** The average fidelity between reconstructed single-photon states $\tau$ and target state $|\psi\rangle_{\gamma,\phi}$ using SLST, self-guided quantum tomography (SGQT), maximum likelihood estimation (MLE) reconstruction. **b** Average fidelity of SLST by increasing experimental runs $M$ from 10 to 1000. **c** Fidelity between reconstructed two-photon states $\tau$ and target state $\rho_\eta$ using SLST and MLE. **d** The fidelities of two-photon states reconstruction from SLST (dash lines) with $M = 2000$ measurements. The solid lines represent the fidelity from MLE tomography with $M = 2000$ measurements. The dots and bars in (**a**) and (**c**) are the mean value and the corresponding standard deviations obtained by repeating the experiment 5 times. The dashed lines and shadings in (**d**) and (**e**) are the mean value and standard deviation obtained by repeating the iteration 5 times.

$|\psi_{\gamma,\phi}\rangle$, we perform shadow tomography and estimate the expectation of $O = |+\rangle\langle+|$. The results of $\max_{\mathbf{P}}\mathrm{Var}(\hat{o})$ are shown in Fig. 2d, in which we observe that $\max_{\mathbf{P}}\mathrm{Var}(\hat{o}^{(m)})$ converges to 0.75 when $M > 255$. In Fig. 2e, we show $\max_{\mathbf{P}}\mathrm{Var}(\hat{o})$ of 128 observables $O \in \mathbf{O}$ with $M = 315$ measurements, which agrees well with the theoretical prediction that the shadow norm is a constant regardless of the explicit form of $O \in \mathbf{O}$[38]. To give a comparison, we simulate $\max_{\mathbf{P}}\mathrm{Var}(\hat{o})$ with symmetric informationally complete (SIC) POVM $\mathbf{E}_{\mathrm{SIC}}$[46], which is constructed with the minimal number of 4 measurements for qubit system and has been widely adopted in investigations of advanced tomography[47–49]. As shown in Fig. 2e, the shadow norm with $\mathbf{E}_{\mathrm{SIC}}$ depends on observable $O$ and generally larger than that with $\mathbf{E}_{\mathrm{octa}}$, which indicates $\mathbf{E}_{\mathrm{octa}}$ requires less shots $M$ than $\mathbf{E}_{\mathrm{SIC}}$ to achieve the same accuracy of estimation $\hat{O}$.

**State reconstruction**

The direct estimation from classical shadows $\hat{\rho}^{(m)}$, i.e., $\hat{\rho} = 1/M\sum_{m=1}^{M}\hat{\rho}^{(m)}$, is generally not a physical state with finite $M$ measurements, which limits the application of shadow tomography in estimation of nonlinear functions[37,50]. Physical constraints need to be introduced to enforce the positivity of the reconstructed state $\tau$, which can be addressed by solving the optimization problem

$$\text{minimize} \quad \hat{N}_F(\tau) = \frac{2}{M(M-1)}\sum_{m<n}\mathrm{Tr}\left[\hat{\rho}^{(m)}\hat{\rho}^{(n)}\right] + \mathrm{Tr}(\tau^2) - 2\sum_m \mathrm{Tr}(\hat{\rho}^{(m)}\tau)$$
$$\text{subject to} \quad \tau \geq 0, \mathrm{Tr}(\tau) = 1, \tag{3}$$

where $\tau$ is the proposed state that is positive semidefinite ($\tau \geq 0$) with unit trace ($\mathrm{Tr}(\tau) = 1$), and the cost function $\hat{N}_F(\tau)$ is the unbiased estimator of squared Frobenius norm with $\{\hat{\rho}^{(m)}\}$ (see Supplementary Note 2A for more details). Note that the squared state fidelity adopted in SGQT[20–22] is not an unbiased estimator with $\{\hat{\rho}^{(m)}\}$ for mixed state. We employ an iterative self-learning algorithm, i.e., SPSA algorithm, to solve the optimization problem in Eq. (3). SPSA is especially efficient in multi-parameter optimization problems in terms of providing a good solution for a relatively small number of measurements of the objective function[51], which holds the similar spirit as shadow tomography. In traditional maximum likelihood estimation (MLE) reconstruction[14], the computational expense required to estimate gradient direction is directly proportional to the number of unknown parameters ($4^N - 1$ for an $N$-qubit state) as it approximates the gradient by varying one parameter at a time, which becomes an issue when the number of qubit is large. In SPSA, the minimization of cost function $\hat{N}_F(\tau)$ is achieved by perturbing all parameters simultaneously, and one gradient evaluation requires only two evaluations of the cost function. While SPSA costs more iterations to converge, it returns state with higher fidelity in

limited number of iterations compared to MLE[21]. More importantly, SPSA formally accommodates noisy measurements of the objective function, which is an important practical concern in experiment.

Generally, an $N$-qubit state $\tau$ can be modeled with $d^2$ parameters with $d = 2^N$ being the dimension of $\tau$. Thus, the proposed state $\tau$ is determined by a $d^2$-dimensional vector $\boldsymbol{r} = [r_1, r_2, \cdots, r_{d^2}]$. SPSA optimization estimates the gradient by simultaneously perturbing all parameters $r_i$ in a random direction, instead of individually addressing each $r_i$. In $k$th iteration, the simultaneous perturbation approximation has all elements of $\boldsymbol{r}_k$ perturbed together by a random perturbation vector $\boldsymbol{\Delta}_k = [\Delta_{k1}, \Delta_{k2}, \cdots, \Delta_{kd^2}]$ with $\Delta_{ki}$ being generated from Bernoulli $\pm 1$ distribution with equal probability. Then the gradient is calculated by

$$\boldsymbol{g}_k = \frac{\hat{N}_F(\boldsymbol{r}_k + B_k\boldsymbol{\Delta}_k) - \hat{N}_F(\boldsymbol{r}_k - B_k\boldsymbol{\Delta}_k)}{2B_k}\boldsymbol{\Delta}_k, \tag{4}$$

and $\boldsymbol{r}_k$ is updated to $\boldsymbol{r}_{k+1}$ by $\boldsymbol{r}_{k+1} = \boldsymbol{r}_k + A_k\boldsymbol{g}_k$. $A_k$ and $B_k$ are functions in forms of $A_k = a_1/(k + a_2)^{a_3}$ and $B_k = b_1/k^{b_2}$ with $a_1, a_2, a_3, b_1$ and $b_2$ being hyperparameters that determine the convergence speed of algorithm, which can be generally obtained from numerical simulations (see Supplementary Note 2B for hyperparameter settings). SLST is terminated when there is little change of $\hat{N}_F(\boldsymbol{r}_k)$ in several successive iterations, and corresponding $\tau_k$ is the reconstructed state. We emphasize that SPSA inevitably introduces systematic errors of the reconstructed state, as well as other optimization algorithms such as MLE and least squares[52]. In fact, it is a tradeoff that the reconstruction of a physical state suffers from a bias.

As the prepared single-photon state is extremely closed to the ideal state $|\psi_{\gamma,\phi}\rangle$, the accuracy of reconstruction is characterized by the state fidelity between returned state $\tau_k$ and ideal state $|\psi_{\gamma,\phi}\rangle$, i.e., $F = \sqrt{\mathrm{Tr}(\tau_k|\psi_{\gamma,\phi}\rangle\langle\psi_{\gamma,\phi}|)}$. The results of average fidelity of SLST over 20 prepared $|\psi_{\gamma,\phi}\rangle \in \mathbf{P}$ after $k = 30$ iterations are shown with red dots in Fig. 3a, where the average fidelity increases as $M$ increases and achieves $0.992 \pm 0.001$ with $M = 315$ measurements. The fabricated metasurface is also capable to collect data required for state reconstruction with other technologies, i.e., SGQT[20–22] and MLE reconstruction (see Supplementary Note 3 for demonstration of SGQT). In SGQT, two projective measurements are performed with 7 experimental runs in each iteration, and SPSA is used to update the proposed state $\tau^{\mathrm{SGQT}}$. The results of $F = \sqrt{\mathrm{Tr}(\tau^{\mathrm{SGQT}}|\psi_{\gamma,\phi}\rangle\langle\psi_{\gamma,\phi}|)}$ are shown with yellow triangles in Fig. 3a, in which we observe an average fidelity of $0.983 \pm 0.003$ after 45 iterations (total experimental runs of 315 as the same as that in SLST). The results of MLE reconstruction $F = \sqrt{\mathrm{Tr}(\tau^{\mathrm{MLE}}|\psi_{\gamma,\phi}\rangle\langle\psi_{\gamma,\phi}|)}$ are

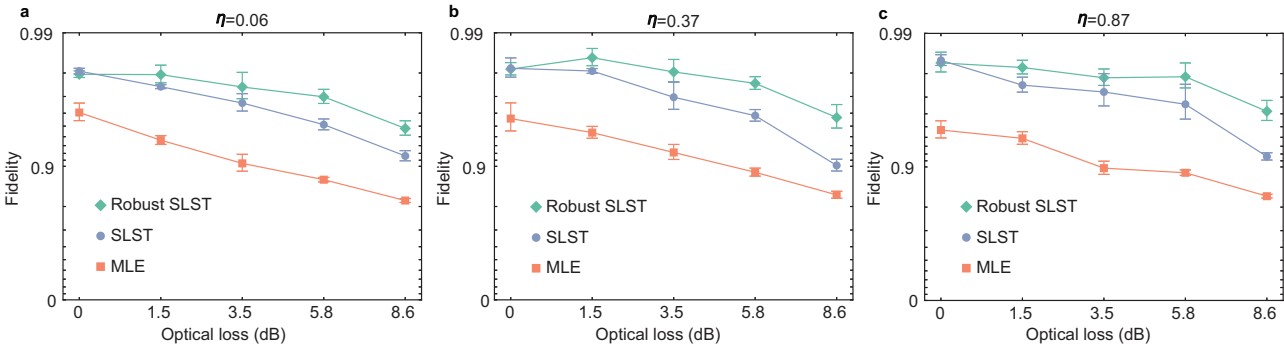

**Fig. 4 | Results of fidelities from robust self-learning shadow tomography (SLST), SLST and maximum likelihood estimation (MLE) reconstruction on two-photon states. a** $\rho_{\eta=0.06}$, **b** $\rho_{\eta=0.37}$ and **c** $\rho_{\eta=0.87}$. In each reconstructions, the experiment is carried out with $M = 1000$ runs. In robust SLST, additional $M' = 2000$ experimental runs are used for calibration. We set $k = 200$ in robust SLST and SLST. The error bars are the standard deviations in SLST (robust SLST), obtained from Monte Carlo simulation with assumption that the collected photons in $M$ ($M'$ and $M$) experimental runs have Poisson distribution.

shown with cyan squares in Fig. 3a. When $M$ is small ($M < 60$), MLE reconstruction is more accurate than SGQT. However, SLST always exhibits higher accuracy compared to other techniques with the same number of experimental runs. It is worth noting that the average fidelity with MLE reconstruction converges to $0.93 \pm 0.01$ and the error of reconstruction is about 0.07, which is consistent with errors in estimation of observables in Fig. 2b. Although the error of metasurface reduces the accuracy of shadow tomography and MLE reconstruction, SLST and SGQT with SPSA optimization can dramatically suppress metasurface-induced error as SPSA can accommodate noisy measurements of the cost function. The accuracy of SLST does not keep increasing with the number of iterations as reflected in Fig. 3b, where the converged fidelity depends on the number of experimental runs $M$ in classical shadow collection.

We also demonstrate SLST on two-photon entangled states $|\psi\rangle_\eta = \sqrt{\eta}|HV\rangle + \sqrt{1-\eta}|VH\rangle$ with $\eta = 0.06, 0.37$ and 0.87. In two-photon SLST, one photon is detected by metasurface-enabled $\mathbf{E}_{octa}$, and the other photon is detected by randomly choosing $\sigma_x$, $\sigma_y$ and $\sigma_z$ measurements realized by an E-HWP and an E-QWP. In contrast to the single-photon state, the generated two-photon state $\rho_\eta$ is far from pure state as it is affected by more noises that are mainly attributed to high-order emission in SPDC and mode mismatch when overlapping two photons in Sagnac interferometer. Thus, the proposed state $\tau_k$ should be a mixed state in general form of $\tau_k = T^\dagger T$ with $T$ being a complex lower triangular matrix (see Supplementary Note 2C for details). Accordingly, the accuracy of reconstruction is characterized by the fidelity between returned state $\tau_k$ and $\rho_\eta$, where $\rho_\eta$ is MLE reconstruction with large amount of data ($M \approx 8 \times 10^5$) collected from bulk optical setting (waveplates and PBS). The results of $F = Tr\left(\sqrt{\sqrt{\tau_{200}}\rho_\eta\sqrt{\tau_{200}}}\right)$ are shown with dots in Fig. 3c, the fidelities of three states reach $0.986 \pm 0.002$, $0.990 \pm 0.001$ and $0.981 \pm 0.002$ with $M = 2000$ experimental runs and $k = 200$ iterations. We also perform MLE reconstruction $\tau^{MLE}$ of two-photon states, where one photon is detected by metasurface and the other is detected by bulk optical setting. The results of $F = Tr\left(\sqrt{\sqrt{\tau^{MLE}}\rho_\eta\sqrt{\tau^{MLE}}}\right)$ with $M$ experimental runs are shown with squares in Fig. 3c. The error in two-photon MLE reconstruction is about $0.047 \pm 0.005$, which is smaller than that in single-photon MLE reconstruction as only one photon is detected by noisy device (metasurface). In Fig. 3d, we show that the fidelity of SLST with $M = 2000$ is converging after $k = 200$ iterations.

### Robust shadow tomography
Finally, we demonstrate robustness of SLST can be further improved by robust shadow tomography[42,53]. Considering that the measurement

apparatus are noisy, the measurement apparatus can be calibrated prior to performing SLST. To this end, shadow tomography is firstly performed on high-fidelity state $|HH\rangle$ with $M'$ experimental runs to calculate the noisy quantum channel $\widetilde{\mathcal{M}}$. Consequently, the classical shadow is constructed by the noisy channel, i.e., $\hat{\rho}^{(m)} = \widetilde{\mathcal{M}}^{-1}(|\psi_{l_1}\rangle\langle\psi_{l_1}| \otimes |\psi_{l_2}\rangle\langle\psi_{l_2}|)$ (See Supplementary Note 1B for details of robust shadow tomography). The framework of robust shadow tomography is valid in our experimental setting. Firstly, although two photons are detected with different measurement devices, i.e., one is the metasurface-enabled POVM while the other is randomly detected on three Pauli bases, the mathematical models of these two measurement devices are identical. Secondly, although the metasurface-induced measurement errors are different between six projections, it has been shown that gate-dependent noise can be suppressed by robust shadow tomography[42]. Finally, the experimental device is able to generate $|HH\rangle$ with sufficiently high fidelity. Otherwise, the noise in state preparation might be added in $\widetilde{\mathcal{M}}$, which introduces biased estimation of returned state. In our experiment, the fidelity of prepared $|HH\rangle$ is $0.9956 \pm 0.0005$ with respect to the ideal form. To demonstrate robust SLST, we insert a tunable attenuator before metasurface to introduce optical loss from 1.5 dB to 8.6 dB, which accordingly reduces the fidelity of prepared state as reflected by the MLE reconstruction shown in Fig. 4. Compared to SLST, robust SLST is able to enhance the accuracy of reconstruction in the presence of optical loss, especially at the high-level optical loss. It is worth mentioning that SLST itself can accommodate metasurface-induced measurement errors so that the enhancement of robust SLST is not significant when optical loss is zero. Increasing the optical loss is equivalent to stronger measurement noise. We observe the significant enhancement of robust SLST at high-level optical loss, which indicates robust SLST can further improve the robustness of SLST against noise (See Supplementary Note 1C for numerical simulations of robust SLST).

## Discussion
We propose and demonstrate POVM with a single metasurface that enables implementation of real-time shadow tomography and observation of sample complexity. Together with the developed SLST, the underlying quantum states can be reconstructed efficiently, accurately and robustly. The advantages are evident even in single- and two-photon polarization-encoded states. The concept of octahedron POVM can be readily realized with integrated optics, where the directional couplers and phase shifters are able to construct octahedron POVM encoded in path degree of freedom. Metasurface-enabled POVM is particularly promising for efficient detection of scalable polarization-encoded multiphoton entanglement, in which two

measurement devices are sufficient for full characterization[54]. Our investigation is compatible with metasurface-enabled generation[7,8] and manipulation[9–11] of photonic states, thereby opening the door to quantum information processing with a single ultra-thin optical device.

## Methods

### Fabrication of metasurface

A 700 nm-thick layer of a-Si is deposited on top of 750 $\mu$m-thick fused quartz wafers using the low-pressure chemical vapor deposition (LPCVD) technique. Then a layer of AR-P6200.09 resists (Allresist GmbH) with a thickness of 200 nm is spun and coated on the substrate. The metasurface pattern is generated with electron-beam lithography (EBL) process which is set with 120 kV, 1 nA current and 300 $\mu c$ cm$^{-2}$ dose. Subsequently, the resist is developed with AR300-546 (Allresist GmbH) for 1 min. Reaction ion etching (RIE) is performed to transfer the nanostructures to a-Si film. The residue resist is removed by immersing the chip first in acetone for 5 min, then in isopropanol for 5 min and finally in deionized water.

### Experimental setup to implement SLST with metasurface

Metasurface is fixed on a piece of hollow plastic, which can be adjusted in six degrees of freedom through a six-dimensional rotation stage. Objective lens with 20 × magnifying factor and tube lens with the focal length of 200 mm is used as a microscope, enlarging the distance of six spots focused by metasurface from 70 $\mu$m to 1.9 mm. Then, three prisms at different heights are applied to separate six light beams. Four mini lenses with $f = 15$ mm and two mini lenses with $f = 30$ mm are used to couple the six beams into six multi-mode fibers with the core diameter of 62.5 $\mu$m.

## Data availability

The data generated in this study have been deposited in the Zenodo database with the identifier https://zenodo.org/records/10674374 [https://doi.org/10.5281/zenodo.10674373].

## Code availability

The codes used for data analysis and simulation in this study have been deposited in the Zenodo database with the identifier https://zenodo.org/records/10674374 [https://doi.org/10.5281/zenodo.10674373].

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

## Acknowledgements

The authors thank Xiaoqi Zhou for insightful discussion. The authors thank the anonymous reviewers for the insightful comments on the work. K. A., T. Z. and H. L. were supported by the National Key Research and Development Program of China (Grant No. 2019YFA0308200), the National Natural Science Foundation of China (Grants Nos. 11974213 and 92065112), Shandong Provincial Natural Science Foundation (Grant Nos. ZR2020JQ05 and ZR2023LLZ005), Taishan Scholar of Shandong Province (Grant No. tsqn202103013), Shenzhen Fundamental Research Program (Grants Nos. JCYJ20190806155211142 and JCYJ20220530141013029), Shandong University Multidisciplinary Research and Innovation Team of Young Scholars (Grant No. 2020QNQT) and Higher Education Discipline Innovation Project ('111') (Grant No. B13029). S. L. and G. W were supported by the National Natural Science Foundation of China (Grants Nos. 62375282 and 62205370). Y. Z. was supported by the National Natural Science Foundation of China (Grants No. 12205048) and Innovation Program for Quantum Science and Technology (Grant No. 2021ZD0302000).

## Author contributions

H.L. conceived and designed the experiment. K.A. T.Z. and H.L. carried out the experiment of SLST. Z.L., S.L., L.W., W.Z. and G.W designed and fabricated metasurface. Y. Z., X. Y. and H. L. conducted the theoretical analysis. H. L. and G. W. supervised the project. All authors contributed to writing the manuscript.

## Competing interests

The authors have no competing interests.
