## [Peer Review File · Nature Communications]

Efficient Characterizations of Multiphoton States with an Ultra-thin Optical DeviceREVIEWER COMMENTS

Reviewer #1 (Remarks to the Author):

The manuscript “Efficient Characterizations of Multiphoton States with Ultra-thin Integrated Photonics” reports an experimental shadow tomography with POVM on photonic states. The POVM is realized by a metasurface, which significantly reduces the experimental complexity compared to bulky optics and improves the accuracy of estimation. Moreover, the authors show that shadow tomography equipped with SPSA optimization and noise calibration enables accurate state reconstruction against optical loss. The technologies are promising for efficient characterization of multiphoton entanglement, and the metasurface-enabled POVM have a sound and positive impact in the emerging field of quantum-photonics applications of metasurface. The paper is well written, and it has been an enjoyable read. I recommend it for publication in Nature Communications.

I have the following comments/suggestions for the authors.

1. The authors state that the metasurface allows for the implementation of real-time shadow tomography. I guess the “real-time” means all information are contained within the single metasurface. Am I correct? Some explanations of “real-time” would be helpful.
2. There is no scale bar in the SEM of metasurface in Figure 1(c).
3. The theoretical shadow norm of octahedron POVM in Phys. Rev. Lett. 129, 220502 (2022). is 1.5, why the value here is 0.75?
4. Why the authors choose SIC POVM for comparison?
5. Why the authors choose fidelity as loss function but not trace distance?
6. The authors demonstrate SGQT on single-photon state, why not implement SGQT on two-photon states as well?
7. The details of implementation of SGQT is missing, and they can go to Supplementary Information.
8. The authors claim that “SLST always exhibits advantage compared to other techniques in presence of experimental noises”. I think “optical loss” is much better here.
9. In page 8, the authors write “This advantage is mainly attributed to the SPSA optimization”. It would be nice if the authors could comment more details.

Reviewer #2 (Remarks to the Author):

The manuscript by An et al introduces an integrated photonics device that promises efficient characterization of polarization-encoded states of single photons. The device effectively acts as a polarizing beam splitter across different axes, which enables the measurement of different qubit mutually unbiased bases in different areas of the devices. The device itself is interesting, although it is questionable whether it should be called integrated optics, given that it requires a free-space beam. Unfortunately, the manuscript mainly focuses on performed tomographies, which distracts from the main achievement, and also raises several questions, as I detailed below. Hence, I believe the paper requires a shift in focus and a revision (or at least a more detailed explanation) of the

tomography results before it can be published.

The metasurface is reminiscent of the early days of encoding information into spatial modes of photons, where Holograms were used for state identification (nowadays replaced by SLMs). Since this is really the main result of the paper, I think much more discussion should be devoted to the metasurface, including a large part of the data from the supplement. A few explicit questions arose:

- For the produced metasurface, the data in the supplement shows still significant measurement errors, particularly in the H/V section of the device. Is the source of this known and what are the prospects of improving the performance to the level of standard waveplates?
- What happens in intermediate regions between the 3 parts of the device?
- Given that the "beam" of photons will generally have a Gaussian profile, how can you achieve equal splitting into the 3 regions of the device?
- The same transformation could be achieved with widely used integrated wave-guide systems. What is the advantage of this device over a standard waveguide-splitter?

Regarding the tomography part, I have several questions and comments:

- Figure 2 needs error bars and it is quite unclear what is really plotted there. I am not sure I understand the physical meaning of what is plotted in Fig 2b. How does this show that one of the POVMs is better than the other? This needs to be explained better.
- On page 6 it is claimed that shadow tomography has limited applicability due to unphysical results. It should be noted that classical shadows converge towards the physical domain very quickly, even for large systems. In contrast, optimization-based methods such as MLE or SPSA hardly scale beyond a handful of qubits to the classical computational complexity.
- Figure 3 presents (as far as I can tell) an unfair comparison. SLST, by construction, seems to find a state that optimizes the fidelity with the observed classical shadow. MLE, on the other hand, finds the state that best explains the observed data, and the fidelity is computed with some target state. From this, we would expect SLST to outperform MLE by construction.
- If one computes the MLE fidelity with the observed shadow, rather than the target state, the whole argument becomes circular and involves an unphysical state, which was to be avoided.
- I was also not able to reproduce the MLE data. I simulated noisy MLE with $M=300$ and over 1000 instances the mean fidelity with the target state was 0.99 with a standard deviation of 1%. This is very much in line with the results for SLST and much better than what Fig 3 shows for MLE.
- On page 8 the authors induce loss and discuss the resulting SNR. I am a bit puzzled by this since two-photon experiments are usually post-selected on coincidence detections, which should hardly be affected by the induced loss.
- The manuscript talks about real-time shadow tomography, but I see no real-time data.

Reviewer #3 (Remarks to the Author):

Brief summary

In the manuscript, An et al. experimentally demonstrate using a single metasurface optical chip for implementing positive-operator-valued measures to perform classical shadow and full quantum state tomography.

With the metasurface optical chip, they realize an octahedral POVM. First, they use this to study the sample complexity of classical shadow tomography estimating expectation values of observables in single photon states. Consistent with theoretical predictions, they find that their octahedral POVM outperforms the SIC POVM. Secondly, they demonstrate full tomographic reconstruction of single and two-photon states introducing a so-called self-learning shadow tomography. This is an iterative algorithm based on simultaneous stochastic perturbation and gradient descent. They find that this algorithm outperforms other techniques, and can be made more robust against experimental noise, using techniques from robust shadow tomography.

Manuscript content

The manuscript addresses a timely and interesting subject – improving efficiency of (classical shadow) tomography using non-trivial POVMs. The metasurface chip offers a novel way to implement such POVMs. With respect to the content, I have the following questions and comments:

- Before studying the sample complexity of classical shadow tomography, it seems logical to first examine the capability to reconstruct desired expectation values accurately. Using the same data from Fig. 2a, it would be beneficial to clearly show that the expectation values of the observables in Fig. 2 are estimated reliably, rather than just focusing on their empirical variance.
- A direct characterization of the POVM is to some extent missing in the main text, or should be more explicitly commented on [as it is the paper's main topic]. For instance, to which fidelity are the projector of 6 states measured? Showing the reconstructed expectation values in Fig. 2 would also serve to characterize the POVM more accurately. While the latter sections on tomography does touch upon this, it's largely within the context of advanced projection and optimization techniques.
- It is unclear to me how the proposed SPSA optimization can be executed with data from randomized measurements if both, theory and experimental state, are mixed. Since in this case, the fidelity F is not a polynomial function of ρ , there doesn't exist an unbiased estimator for F using classical shadows. Is the presented algorithm thus specific to pure theory states [in which case the fidelity is a simple overlap] - if so this should be stated clearly – or can it be used for mixed theory states as well?
- The two photon states are not defined. Why are not both photons detected with a metasurface POVM? Connected to this, how is the scalability to detect many-photon states? Does one need a separate metasurface for each photon?
- The robust shadow estimation should be explained in more detail, in particular since the two-photon states are detected with two different detectors. Also, robust shadow estimation makes certain assumptions about the noise. Are these fulfilled?

Manuscript presentation

The manuscript is overall well-written and well-structured. However, details and proper definitions are sometimes missing or can only be guessed from context. This concerns for instance (but not only):

- Throughout the manuscript, one should carefully distinguish between shadow tomography (Aaronson et al.) and classical shadow tomography (Huang et al.).
- Page 3, first paragraph: Should it read “Proportional to the projector to ...”
- Page 3, first paragraph: It would be helpful to explain in more details why every 2-design gives rise to a POVM. A 1-design should be sufficient?
- Page 4, last paragraph: O and P are not defined.
- Page 5: How is the maximum over P computed?
- Page 5: SIC POVM is not defined.

Conclusion

Once the above points have been addressed and the presentation has been improved, the manuscript should be reconsidered for publication in Nature Communications.

Summary of Changes

In response to Reviewer 1's comments and suggestions, we have made the following changes.

1. To address Comment 1, we have added a sentence in the second paragraph on Page 2 and added data in Figure 2(c).
2. To address Comment 2, we have revised Figure 1(c).
3. To address Comment 3, we have added a section (I C) in Supplementary Materials.
4. To address Comment 4, we have added a sentence in the first paragraph on Page 7.
5. To address Comment 5, we have revised the part regarding loss function on Page 7.
6. To address Comments 6 and 7, we have added a section (III) in Supplementary Materials.
7. To address Comment 8, we have revised "experimental noise" to "optical loss".
8. To address Comment 9, we have revised the end of second paragraph on page 7.

In response to Reviewer 2's comments and suggestions, we have made the following changes.

1. We have revised "integrated optics" to "optical device".
2. To address Comments 1 and 2, we have revised the first paragraph on Page 5, added results in Figure 1(e) and (f) and significantly revised Supplementary Materials IV B.
3. To address Comment 3, we have revised first paragraph on Page 4 and added a section (V B) in Supplementary Materials.
4. To address Comment 4, we have revised the conclusion on Page 10.
5. To address Comment 5, we have revised Figure 2 along with explicit descriptions from Page 5 to Page 9.
6. To address Comments 6, 7, 8 and 9, we have made revisions from Page 5 to Page 9 and added a section (V C) in Supplementary Materials.
7. To address Comment 10, we have revised the first paragraph on page 10.
8. To address Comment 11, we have added results in Figure 2(c).

In response to Reviewer 3's comments and suggestions, we have made the following changes.

1. To address Comments 1 and 2, we have revised the first and second paragraphs on Page 5, added results in Figure 1(e), (f) and Figure 2(b).
2. To address Comment 3, we have revised the part regarding loss function on Page 7 and added a section (II A) in Supplementary Materials.
3. To address Comment 4, we have revised the second paragraph on page 9 and conclusion.
4. To address Comment 5, we have revised the first paragraph on Page 10 and Supplementary Materials I B.

5. To address Comment 6, we have revised the second paragraph on Page 2.
6. To address Comments 7 and 8, we have revised the second paragraph on Page 3
7. To address Comment 9, we have revised the second paragraph on Page 5.
8. To address Comment 10, we have revised the first paragraph on Page 6.
9. To address Comment 11, we have revised the first paragraph on page 7 and Supplementary Materials I D.

We have revised the main text and Supplementary Materials according to all the comments and suggestions raised by 3 Reviewers, and all the changes have been highlighted in blue. Below, please find our point-to-point responses (in black) to Reviewers' reports (in blue).

Reply to the Report of Reviewer 1

The manuscript “Efficient Characterizations of Multiphoton States with Ultra-thin Integrated Photonics” reports an experimental shadow tomography with POVM on photonic states. The POVM is realized by a metasurface, which significantly reduces the experimental complexity compared to bulky optics and improves the accuracy of estimation. Moreover, the authors show that shadow tomography equipped with SPSA optimization and noise calibration enables accurate state reconstruction against optical loss. The technologies are promising for efficient characterization of multiphoton entanglement, and the metasurface-enabled POVM have a sound and positive impact in the emerging field of quantum-photonics applications of metasurface. The paper is well written, and it has been an enjoyable read. I recommend it for publication in Nature Communications.

I have the following comments/suggestions for the authors.

We appreciate the time and effort Reviewer took to review our manuscript, and we thank Reviewer for the insightful comments that help us to improve the manuscript. In particular, her/his comment on the choice of loss function in SLST helps us to refine the algorithm with unbiased estimator. Below are the point-to-point responses to Reviewer’s comments.

Comment 1: The authors state that the metasurface allows for the implementation of real-time shadow tomography. I guess the “real-time” means all information are contained within the single metasurface. Am I correct? Some explanations of “real-time” would be helpful.

Reply: We appreciate Reviewer for this valuable remark. We agree with Reviewer’s understanding regarding “real-time”, and we are sorry that explicit explanation of “real-time” is missing in previous manuscript. In shadow tomography (or randomized measurement schemes), the information-complete measurements are required for each experimental run (single-shot experiment), which generally requires of different measurement settings. The metasurface-enabled POVM \mathbf{E}_{octa} we demonstrated here is information-(over)complete so that the complete information for tomography is contained in every single experimental run. This distinct feature allows an experimentalist free to stop shadow tomography at any point.

In the revision, we have clarify “real-time”. Moreover, we have revised Figure 2(c) to show the real-time estimation of expectations of observables, which is also attached below in Figure 1

Comment 2: There is no scale bar in the SEM of metasurface in Figure 1(c).

Reply: We thank Reviewer for this remark, and we have added the scale bar in Figure 1(c) in revision.

Comment 3: The theoretical shadow norm of octahedron POVM in Phys. Rev. Lett. 129, 220502 (2022), is 1.5, why the value here is 0.75?

Reply: We thank Reviewer for this remark. For a given POVM \mathbf{E} , the shadow norm of observable O is derived from the variance of the estimation \hat{o} on state ρ . The variance $\text{Var}(\hat{o})$ can be ideally written as

$$\text{Var}(\hat{o}) = \sum_{l=1}^L \text{Tr}(\hat{\rho}_l O)^2 \text{Tr}(\rho E_l) - \text{Tr}(\rho O)^2. \quad (1)$$

Figure 1: Real-time estimation of expectations of five observables O , which are randomly selected from set \mathbf{O} .

The shadow norm is then defined by the maximization of variance $\text{Var}(\hat{\rho})$ over ρ

$$\text{Var}(\hat{\rho}) = \sum_{l=1}^L \text{Tr}(\hat{\rho}_l O)^2 \text{Tr}(\rho E_l) - \text{Tr}(\rho O)^2 \leq \max_{\rho} \sum_{l=1}^L \text{Tr}(\hat{\rho}_l O)^2 \text{Tr}(\rho E_l) - \text{Tr}(\rho O)^2. \quad (2)$$

Note that the second term $\text{Tr}(\rho O)^2$ is a constant and can be ignored. Then, the shadow norm of O is calculated by

$$\|O\|_{\text{shd}}^2 = \lambda_{\max} \left\{ \sum_{l=1}^L \text{Tr}(\hat{\rho}_l O)^2 E_l \right\}, \quad (3)$$

with $\lambda_{\max} \{ \cdot \}$ being the maximal eigenvalue of corresponding operator. In theoretical investigations such as [Phys. Rev. Lett. 129, 220502 (2022)], it is convenient to calculate shadow norm in Eq. 3. For octahedron POVM, the theoretical shadow norm calculated according to Eq. 3 is $\|O\|_{\text{shd}}^2 = 1.5$.

Experimentally, the variance of estimator $\hat{\rho}$ is observed by

$$\text{Var}(\hat{\rho}) = \frac{1}{M} \sum_{m=1}^M \left(\hat{\rho}^{(m)} - \hat{\rho} \right)^2. \quad (4)$$

Without consideration of experimental noise, Eq. 4 converges to Eq. 1 when $M \rightarrow \infty$. For octahedron POVM, the maximization of Eq. 1 over single-qubit pure state ρ yields $\|O\|_{\text{shd}}^2 = 0.75$, which is considered as the theoretical prediction for experimentally observed variance in Eq. 4.

In the revision, we have added derivation in Supplementary Materials to clarify this point.

Comment 4: Why the authors choose SIC POVM for comparison?

Reply: We thank Reviewer for this remark. In quantum state tomography (QST), an information-complete set of measurements is required to reconstruct the complete description of quantum state. SIC POVM has been proved to be the optimal measurement as it is constructed with the minimal number of d^2 measurements for a d -dimensional state. Along this spirit, SIC POVM is widely adopted in the investigations of advanced

tomography schemes, such as adaptive tomography [PRX Quantum 2, 040342 (2021)] and neural network QST [Nat. Phys. 14, 447 (2018); Nat. Machine Intell. 1, 155 (2019)], just name a few.

However, SIC POVM does not always exhibits advantage. Theoretical studies [Phys. Rev. Lett. 129, 220502 (2022)] have shown that octahedron POVM outperforms SIC POVM in shadow tomography in terms of shadow norm (sample complexity), i.e., the shadow norm with octahedron POVM is smaller than that with SIC POVM. In our experiment, we design and fabricate the octahedron POVM device (metasurface), and observe the shadow norm. Furthermore, we compare the experimental shadow norm with octahedron POVM and simulated shadow norm with SIC POVM (ideal results without consideration of noise), which strongly convinces the theoretical prediction of the advantage of octahedron POVM.

In the revision, we have clarified the significance of SIC POVM.

Comment 5: Why the authors choose fidelity as loss function but not trace distance?

Reply: We thank Reviewer for this insightful remark. Indeed, the fidelity $\hat{F}(\tau)$ is an *unbiased* estimator with classical shadows $\{\hat{\rho}^{(m)}\}$ if τ is pure state. However, $\hat{F}(\tau)$ given in the previous version is not a polynomial function of $\hat{\rho}$ for a mixed state τ , which accordingly makes $\hat{F}(\tau)$ *not* an unbiased estimator with $\{\hat{\rho}^{(m)}\}$. Similarly, trace distance is not an unbiased estimator with $\{\hat{\rho}^{(m)}\}$.

In the revision, we address this issue by using the squared Frobenius norm between τ and ρ

$$N_F(\tau, \rho) = \|\rho - \tau\|_F^2 = \text{Tr}(\rho - \tau)^2 = \text{Tr}(\rho^2) + \text{Tr}(\tau^2) - 2\text{Tr}(\rho\tau), \quad (5)$$

as loss function in SPSA. For this cost function, we indeed can write down the *unbiased* estimator with shadows $\{\hat{\rho}^{(m)}\}$ as follows.

For the first term, it shows

$$\frac{2}{M(M-1)} \sum_{m < n} \text{Tr} [\hat{\rho}^{(m)} \hat{\rho}^{(n)}], \quad (6)$$

which is an order-2 polynomial function of $\hat{\rho}^{(m)}$ [Nat. Phys. 16, 1050 (2020)]. Obviously, $\text{Tr}(\hat{\rho}\tau)$ is an *unbiased* estimator as well. Then, the *unbiased* estimator of N_F in total is

$$\hat{N}_F(\tau) = \frac{2}{M(M-1)} \sum_{m < n} \text{Tr} [\hat{\rho}^{(m)} \hat{\rho}^{(n)}] + \text{Tr}(\tau^2) - 2 \sum_m \text{Tr}(\hat{\rho}^{(m)}\tau). \quad (7)$$

Accordingly, the gradient is changed to

$$\mathbf{g}_k = \frac{\hat{N}_F(\mathbf{r}_k + \beta_k \mathbf{\Delta}_k) - \hat{N}_F(\mathbf{r}_k - \beta_k \mathbf{\Delta}_k)}{2\beta_k} \mathbf{\Delta}_k. \quad (8)$$

In the revision, we have updated the results of SLST using the squared Frobenius norm $\hat{N}_F(\tau)$ as cost function.

Comment 6: The authors demonstrate SGQT on single-photon state, why not implement SGQT on two-photon states as well?

Reply: We thank Reviewer for this remark. In single-photon experiment, SGQT requires projective measurement on arbitrary single-qubit pure state, which can be realized by an E-HWP, an E-QWP and metasurface with our setup. However, in two-photon experiment, SGQT requires the projective measurement on arbitrary

two-qubit states including the entangled states, which is still challenging with linear optics. Although two-photon SGQT can be effectively demonstrated with local measurements and post-processing, the advantages of SGQT is accordingly limited.

Comment 7: The details of implementation of SGQT is missing, and they can go to Supplementary Information.

Reply: We thank Reviewer for this remark. In the revision, we have added a section to detail the implementation of single-photon SGQT in Supplementary Materials.

Comment 8: The authors claim that “SLST always exhibits advantage compared to other techniques in presence of experimental noises”. I think “optical loss” is much better here.

Reply: We thank Reviewer for this remark, and we have accordingly revised manuscript.

Comment 9: In page 8, the authors write “This advantage is mainly attributed to the SPSA optimization”. It would be nice if the authors could comment more details.

Reply: We thank Reviewer for this insightful remark and we agree with Reviewer this should be clarified more clearly. Because of the metasurface-induced errors in POVM, the accuracy of estimations with classical shadows is about 0.07 as reflected by experimental results of estimated expectations of observables. However, this error can be dramatically eliminated in SLST due to the SPSA optimization. SPSA is especially efficient in high-dimensional problems in terms of providing a good solution for a relatively small number of measurements of the objective function, which holds the similar spirit as shadow tomography. More importantly, SPSA formally accommodates noisy measurements of the objective function, which is an important practical concern in experiment.

In the revision, we have made this statement more clearly.

Reply to the Report of Reviewer 2

The manuscript by An et al introduces an integrated photonics device that promises efficient characterization of polarization-encoded states of single photons. The device effectively acts as a polarizing beam splitter across different axes, which enables the measurement of different qubit mutually unbiased bases in different areas of the devices. The device itself is interesting, although it is questionable whether it should be called integrated optics, given that it requires a free-space beam. Unfortunately, the manuscript mainly focuses on performed tomographies, which distracts from the main achievement, and also raises several questions, as I detailed below. Hence, I believe the paper requires a shift in focus and a revision (or at least a more detailed explanation) of the tomography results before it can be published. The metasurface is reminiscent of the early days of encoding information into spatial modes of photons, where Holograms were used for state identification (nowadays replaced by SLMs). Since this is really the main result of the paper, I think much more discussion should be devoted to the metasurface, including a large part of the data from the supplement. A few explicit questions arose:

We appreciate the time and efforts Reviewer took to review our manuscript, and we thank Reviewer for her/his insightful comments and suggestions. Her/his comments help us to investigate metasurface along the line that we overlooked in previous manuscript, which is significantly related to the performance of shadow tomography. We agree with Reviewer that “it is questionable whether it should be called integrated optics”, which has been changed to “optical device” in revision. Below are the point-to-point responses to Reviewer’s comments.

Comment 1: For the produced metasurface, the data in the supplement still shows significant measurement errors, particularly in the H/V section of the device. Is the source of this known and what are the prospects of improving the performance to the level of standard waveplates?

Reply: We appreciate Reviewer for this insightful comment, which helps us to thoroughly analyse how the errors in metasurface influence different tomographic technologies. First of all, we apologize for the mistake that the reconstructed Stokes parameters in Figure 4 and Figure 5 in Supplementary Materials are both from numerical simulation. We have reanalyzed data, and the results are listed in Table 1. Compared with the ideal value, the average errors of s_1 , s_2 and s_3 are 0.097, 0.027 and 0.029 in simulation and 0.101 ± 0.005 , 0.086 ± 0.005 and 0.073 ± 0.005 in experiment.

Stokes parameters	s_1		s_2		s_3	
	Sim.	Exp.	Sim.	Exp.	Sim.	Exp.
$ H\rangle$	0.968	0.996 ± 0.001	0.025	-0.100 ± 0.016	0.024	-0.025 ± 0.017
$ V\rangle$	-0.991	-0.913 ± 0.003	-0.046	0.124 ± 0.016	-0.023	0.083 ± 0.016
$ +\rangle$	-0.120	-0.132 ± 0.016	0.988	0.936 ± 0.002	0.061	0.181 ± 0.016
$ -\rangle$	-0.140	0.148 ± 0.016	-0.986	-0.925 ± 0.002	-0.046	0.064 ± 0.016
$ R\rangle$	-0.147	0.163 ± 0.016	0.031	-0.122 ± 0.016	0.991	0.920 ± 0.003
$ L\rangle$	-0.133	-0.069 ± 0.016	-0.033	-0.034 ± 0.017	-0.992	-0.995 ± 0.001

Table 1: The simulation and experimental results of reconstructed Stokes parameters.

Consequently, the metasurface-induced measurement errors reduce the accuracy of shadow tomography and MLE reconstruction as reflected by experimental results. However, we would like to clarify that the SPSSA algorithm in SLST dramatically eliminate the metasurface-induced errors in state reconstruction, and this is actually the advantage of our work.

1, Is the source of this known?

The sources of polarization measurement errors with metasurface are mainly attributed to the limitations in design of metasurface and the imperfections in fabrication of metasurface. Let us elaborate.

- 1) **Limitations in design.** The key ingredient in the design of metasurface is to discretize the phase front. According to requirement of deflecting individual polarization to desired direction, we calculate a discrete phase accumulation of metasurface. Then, a periodic array of nanopillars is designed to realize such a discrete phase accumulation. However, the main limitation in such design is discretization itself, which inevitably introduces polarization measurement errors in our case. Moreover, the cross-polarization effect employed to separate circular polarizations ($|R\rangle$ and $|L\rangle$) introduces errors as well.
 - (a) The discretization of phase front is an approximation of its continuous counterpart (bulk optics), which limits the accuracy in phase modulation. Consequently, the metasurface to deflect input polarization (for example $|H\rangle$) to desired direction cannot completely block its orthogonal polarization ($|V\rangle$) transmitting along the same direction, which introduces polarization measurement error.
 - (b) The metasurface to realize discrete phase modulation can be considered as a grating. For large bending angles (43° in our design), it inevitably deflects the incident polarization into other grating orders (undesired directions) [Nano. Lett. 17, 6267 (2017)].
 - (c) Particularly, to separate circular polarizations, i.e., $|R\rangle$ and $|L\rangle$, the cross-polarization effect has been employed in the design of metasurface, in which the conversion between two orthogonal circular polarizations ($|R\rangle \rightarrow |L\rangle, |L\rangle \rightarrow |R\rangle$) is firstly taken place on the metasurface. However, it is still challenging to realize complete conversion between $|R\rangle$ and $|L\rangle$ on metasurface. Specifically, the aspect ratio constraints of nanopillars, adjacent coupling between nanopillars as well as material absorption would reduce the efficiency of conversion, which consequently increases the errors in polarization measurement.
- 2) **Imperfections in fabrication.** The imperfections in fabrication process of metasurface, such as side-wall smoothness, film thickness, surface roughness and etching residue, affect the efficacy of the metasurface and increase the error in polarization measurement.

2, The significant measurement errors, particularly in the H/V section of the device.

The significant measurement errors in H/V section is mainly attributed to the asymmetric response of H/V section with input polarization of $|H\rangle$ and $|V\rangle$. We simulate the optical response of region 1 (H/V section) with input polarization of $|H\rangle$ and $|V\rangle$, respectively. The simulation is performed within range $x \in [-3\mu m, 3\mu m]$ and $y \in [-1\mu m, 1\mu m]$, which is scaling-down of region 1. As shown in Figure 2(a), the optical responses with input polarization of $|H\rangle$ and $|V\rangle$ are asymmetric with respect of y axis, leading to different transmit efficiency on focal plane. This is verified by simulation of distribution of power intensity on focal plane with input polarization of $|+\rangle = \frac{1}{\sqrt{2}}(|H\rangle + |V\rangle)$ as shown in Figure 2(d), where the output intensities are unbalanced. This unbalanced splitting introduces more errors in reconstruction of $s_1 = \frac{I_H - I_V}{I_H + I_V}$.

In contrast to region 1, region 2 ($|+\rangle/|-\rangle$ section) and region 3 ($|R\rangle/|L\rangle$ section) response their input polarization in symmetric manner. As shown in Figure 2(b) and (c), the distributions of response intensity with

Figure 2: **Simulation of optical response of metasurface with different input polarizations.** Optical response with input polarization of (a), $|H\rangle$ and $|V\rangle$, (b), $|+\rangle$ and $|-\rangle$, (c), $|R\rangle$ and $|L\rangle$. The distribution of power intensity on focal plane with input polarization of (d), $|+\rangle$, (e), $|H\rangle$ and (f), $|H\rangle$.

input polarization of $|+\rangle$ ($|R\rangle$) and $|-\rangle$ ($|L\rangle$) are symmetric with respect to $x = 0$, which consequently leads the balanced splitting of power intensity of input polarization $|H\rangle$ as shown in Figure 2(e) and (f). Therefore, the errors in reconstruction of $s_2 = \frac{I_+ - I_-}{I_+ + I_-}$ and $s_3 = \frac{I_R - I_L}{I_R + I_L}$ are smaller than that of s_1 .

3, what are the prospects of improving the performance to the level of standard waveplates?

It is still challenging to improve the performance of metasurface to the level of standard waveplates. We would like to emphasize that the self-learning shadow tomography (SLST) could dramatically eliminate the measurement errors introduced by metasurface, resulting in the reconstructed state with high accuracy comparable to the results obtained with bulk optics. Indeed, there are several schemes and techniques can improve the performance of metasurface, which are listed below.

- 1) **High-order diffraction suppression.** The high-order diffraction caused by the beam deflection can be suppressed by design of metasurface. For example, asymmetric grating profile [Nano. Lett. 17, 6267 (2017)] and nonperiodic metagrating designs [Nat. Commun. 14, 5602 (2023)] have been proposed to suppress high-order diffraction and deflect the light into a single desired order.
- 2) **Multi-layer metasurface.** Stacking multiple layers of metasurface with varied polarization filtering functionalities is able to enhance the accuracy of polarization control. For example, utilizing the double-layer chiral metasurface [Light: Sci. Appl. 8, 78 (2019)], the average measurement errors of Stokes parameters s_1 , s_2 , and s_3 were achieved at near-infrared wavelengths of 1.9%, 2.7% and 7.2%, respectively. The corresponding results in our work are $10.1\% \pm 0.5\%$, $8.6\% \pm 0.5\%$ and $7.3\% \pm 0.5\%$.
- 3) **Calibration.** For general applications, the calibration process could eliminate errors introduced by inhomogeneity of the incident light shed on metasurface, including power intensity and incident angle [Appl. Sci. 8, 4 (2018)]. Particularly, it has been proved that after calibration, the performance of metasurface-enabled polarimeter is comparable to that of bulk optics [Appl. Opt. 62, 1704 (2023)].

In the revision, we have significantly revised main text as well as Supplementary materials to clarify the influence of metasurface-induced errors in conventional tomography and SLST.

Comment 2: What happens in intermediate regions between the 3 parts of the device?

Reply: We thank Reviewer for this Remark. In fact, there is no specially designed gap between two neighboring regions, i.e., the distance between the marginal nanopillars of neighboring regions is 500 nm, which is as the same as the distance between the other nanopillars. Such a design would introduce a “cut-off” in phase configuration on metasurface, but would not significantly affect the deflection of polarizations. To verify this point, we have simulated the phase configuration on metasurface with input polarization of $|V\rangle$. As shown in Figure 3(a), there are two cut-off lines between three regions, which introduces undesired scattering and consequently increases the optical loss. However, this optical loss is small as most of the incident light is focused around the desired spot as shown in Figure. 3(b) and (c), which is then coupled into optical fibers in our experiment.

In the revision, we have added the discussion regarding this point in Supplementary Information.

Figure 3: **Simulation of phase configuration and distribution of intensity on focal plane with input polarization of $|V\rangle$.** (a), The phase distribution on metasurface. (b), The distribution of intensity on the focal plane. (c), The distribution of power intensity along the dashed lines in (b).

Comment 3: Given that the “beam” of photons will generally have a Gaussian profile, how can you achieve equal splitting into the 3 regions of the device?

Reply: We thank Reviewer for this insightful remark, and we are sorry for inappropriate description. In fact, we are not able to equally split power intensity of input light into three regions. Instead, we post select the photons passing through each region with equal probability. Let us elaborate.

First, the beam waist of the input light is carefully adjusted to be $w_0 = \sqrt{2} \times 210\mu\text{m}$ with lens, which enables the maximal overlap between beam waist and metasurface ($210\mu\text{m}$ square). Then, we carefully locate the metasurface at the center of beam waist, which enables the equal probability of a single photon passing through region 1 ($|H\rangle / |V\rangle$ basis) and region 3 ($|R\rangle / |L\rangle$ basis), i.e., the count rates of collected photons passing through these two regions are the same. However, the single photon passes through region 2 ($|+\rangle / |-\rangle$ basis) with higher probability due to the nature of Gaussian distribution. In our experiment, we randomly discard the collected photons passing through region 2 to make the count rate equal to that of region 1 and region 3. Such experimental setting enables the equal probability of single photon passing through three regions.

Indeed, there is photon loss in this “post-selection” approach. We would like to emphasize that this issue can be addressed with other techniques listed below.

1. **Center symmetrical division of three regions.** The three regions can be divided symmetrically, which avoids the unbalanced splitting of power intensity with input Gaussian beam. For example, the symmetrical division of squared metasurface has been reported in [Laser Photonics Rev. 14, 2000116 (2020)]. Also, the hexagonal metasurface with center symmetrical division of three regions in [Adv. Photon. Res. 3, 2100373 (2022)] is a better option as the hexagonal design enhances the overlap between metasurface and beam waist.
2. **Reshaping the Gaussian beam.** Commercially available diffractive optical devices, such as flat-top beam shapers, are able to transform the Gaussian beam into squared beam where the power intensity is uniformly distributed. Such transformation can also be realized by modulating the phase front of incident light with a liquid crystal spatial light modulator. The squared beam maximizes the overlap with squared metasurface, and the uniformity alleviates the requirement of center symmetrical division of three regions.

In the revision, we have clarified this point and accordingly discussed the solutions.

Comment 4: The same transformation could be achieved with widely used integrated wave-guide systems. What is the advantage of this device over a standard waveguide-splitter?

Reply: We thank Reviewer for this remark. The aim of our work is to efficiently detect photonic states that are encoded in polarization degree of freedom (DOF). Recent development has shown the scalability to generate large-scale multiphoton entanglement encoded in polarization DOF with the assistant of time multiplexing [Phys. Rev. Lett. 129, 150501 (2022)]. Our investigation provides an efficient detection scheme for such large-scale multiphoton entanglement with a *single and compact* device.

In integrated wave-guide systems, the most popular DOF to encode a photon as a qubit is path DOF, and remarkable progresses have been achieved along this line of research. Indeed, POVM applied on photons encoded in path DOF can be readily realized with waveguide-splitters and phase shifters on a *single* chip. However, for the polarization-encoded photons, it is still challenging—to the best of our knowledge—to realize POVM with a *single* optical device.

Both technologies are promising for generation of large-scale multiphoton states, aiming powerful quantum computation. In particular, for the sequentially (time multiplexing) generated multiphoton entanglement, two measurement devices are sufficient for full characterization. Along this spirit, the metasurface we reported is promising to efficiently detect large-scale multiphoton entanglement.

In the revision, we have clarified this point.

Regarding the tomography part, I have several questions and comments:

Comment 5: Figure 2 needs error bars and it is quite unclear what is really plotted there. I am not sure I understand the physical meaning of what is plotted in Fig 2b. How does this show that one of the POVMs is better than the other? This needs to be explained better.

Reply: We thank Reviewer for this remark, and we are sorry for the ambiguous statements in previous manuscript.

The theoretical investigation [Phys. Rev. Lett. 129, 220502 (2022)] has drawn the conclusion that octahedron POVM is optimal in estimation of linear observables with shadow tomography in terms of shadow norm, which determines the sample complexity in shadow tomography. For a given POVM \mathbf{E} , the shadow norm of observable O is derived from the variance of the estimation \hat{o} on state ρ . The variance $\text{Var}(\hat{o})$ can be ideally written as

$$\text{Var}(\hat{o}) = \sum_{l=1}^L \text{Tr}(\hat{\rho}_l O)^2 \text{Tr}(\rho E_l) - \text{Tr}(\rho O)^2. \quad (9)$$

The shadow norm is then defined by the maximization of variance $\text{Var}(\hat{o})$ over ρ

$$\text{Var}(\hat{o}) = \sum_{l=1}^L \text{Tr}(\hat{\rho}_l O)^2 \text{Tr}(\rho E_l) - \text{Tr}(\rho O)^2 \leq \max_{\rho} \sum_{l=1}^L \text{Tr}(\hat{\rho}_l O)^2 \text{Tr}(\rho E_l) - \text{Tr}(\rho O)^2. \quad (10)$$

Note that the second term $\text{Tr}(\rho O)^2$ is a constant and can be ignored. Then, the shadow norm of O is calculated by

$$\|O\|_{\text{shd}}^2 = \lambda_{\max} \left\{ \sum_{l=1}^L \text{Tr}(\hat{\rho}_l O)^2 E_l \right\}, \quad (11)$$

with $\lambda_{\max} \{ \cdot \}$ being the maximal eigenvalue of corresponding operator. In theoretical investigations such as [Phys. Rev. Lett. 129, 220502 (2022)], it is convenient to calculate shadow norm in Eq. 11. For octahedron POVM, the theoretical shadow norm calculated according to Eq. 11 is $\|O\|_{\text{shd}}^2 = 1.5$.

Experimentally, the variance of estimator \hat{o} is observed by

$$\text{Var}(\hat{o}) = \frac{1}{M} \sum_{m=1}^M \left(\hat{o}^{(m)} - \hat{O} \right)^2. \quad (12)$$

Without consideration of experimental noise, Eq. 12 converges to Eq. 9 when $M \rightarrow \infty$. For octahedron POVM, the maximization of Eq. 9 over single-qubit pure states ρ yields $\|O\|_{\text{shd}}^2 = 0.75$, which is considered as the theoretical prediction for experimentally observed variance in Eq. 12. In experiment, we prepare 20 $|\psi\rangle_{\gamma, \phi}$ that are uniformly distributed on the Bloch sphere, forming a set of \mathbf{P} . Then, for an observable O , the experimentally obtained shadow norm is $\max_{\mathbf{P}} \text{Var}(\hat{o})$.

In Figure 2(a) (Figure 2(d) in the revision), we show that the observed shadow norm converges to its theoretical value with $M \approx 300$ shots. In Figure 2(b) (Figure 2(e) in the revision), we show the shadow norm is independent with observables O . To give a comparison, we simulate the results of shadow norm with SIC POVM. The simulation is performed with 315 runs, which is as the same number of experimental runs as we demonstrate in experiment. The shadow norm with SIC POVM is larger than that with octahedron POVM, which indicates SIC POVM requires more shots than octahedron POVM to estimate an observable with the same accuracy.

In the revision, we have made a detailed explanation and added errors in Figure 2.

Comment 6: On page 6 it is claimed that shadow tomography has limited applicability due to unphysical results. It should be noted that classical shadows converge towards the physical domain very quickly, even for large systems. In contrast, optimization-based methods such as MLE or SPSA hardly scale beyond a handful of qubits to the classical computational complexity.

Reply: We thank Reviewer for this remark. Indeed, shadow tomography allows one to infer many properties of a quantum state, including some nonlinear observables such as purity and Rényi entropy. However, shadow tomography is not able to easily predict the properties that cannot be expressed in terms of directly observable (polynomial) quantities such as von Neumann entropy $S(\rho) = -\text{Tr}(\rho \log \rho)$, which has been found widespread applications in quantum state quantification [Science 364, 260 (2019)], quantum thermodynamics [Phys. Rev. Lett. 111, 250404 (2013)], the characterization of topological matter [Phys. Rev. Lett. 96, 110404 (2006); Science 374, 1237 (2021)] and dynamics out of equilibrium [Phys. Rev. Lett. 110, 260403 (2013)]. Instead, our solution is to reconstruct a physical state with the *same set* of collected data from shadow tomography, and then the quantities of interest, such as von Neumann entropy, can be directly calculated with the reconstructed state.

We agree with Reviewer that “optimization-based methods such as MLE or SPSA hardly scale beyond a handful of qubits to the classical computational complexity.” SPSA is unlikely to be the ultimate solution. However, SPSA is especially efficient in high-dimensional problems in terms of providing a good solution for a relatively small number of measurements of the objective function, which holds the similar spirit as shadow tomography. More importantly, SPSA formally accommodates noisy measurements of the objective function, which is an important practical concern in experiment.

The proposed SLST inherits the advantages of shadow tomography and SPSA, i.e., relatively small number of measurements and resilience to practical noise. We believe SLST would benefit intermediate near term quantum optics, and we expect the more efficient optimization algorithms to boost shadow tomography in future.

In the revision, we have clarified the advantage as well as limitations of SLST.

Comment 7: Figure 3 presents (as far as I can tell) an unfair comparison. SLST, by construction, seems to find a state that optimizes the fidelity with the observed classical shadow. MLE, on the other hand, finds the state that best explains the observed data, and the fidelity is computed with some target state. From this, we would expect SLST to outperform MLE by construction.

Reply: We thank Reviewer for this insightful remark, and we are sorry that we leave Reviewer the impression that we are making unfair comparison.

In fact, the comparison between SLST and MLE is fair. Either SLST or MLE reconstructs a physical state τ with the *same* number of experimental shots. The comparison is then taken place by calculating the fidelity between reconstructed state τ and target state. In single-photon experiment, the target state is pure state $\rho_{\gamma,\phi} = |\psi_{\gamma,\phi}\rangle \langle\psi_{\gamma,\phi}|$ because the high fidelity of the prepared single-photon state. In two-photon experiment, the target state ρ_{η} is MLE reconstruction from large amount of experimental runs ($M \approx 8 \times 10^5$) with bulk optics.

In SLST, the fidelity between τ and shadow estimation $\hat{\rho}$ is calculated as loss function, which may leave Reviewer the impression that the results shown in Figure 3 are $F(\hat{\rho}, \tau)$. In fact, Figure 3 (a) and (b) represent the results of $F(\rho_{\gamma,\phi}, \tau)$, and Figure 3 (c) and (d) represent the results of $F(\rho_{\eta}, \tau)$.

In the revision, we have made the statement much more clearly.

Comment 8: If one computes the MLE fidelity with the observed shadow, rather than the target state, the whole argument becomes circular and involves an unphysical state, which was to be avoided.

Reply: We thank Reviewer for this remark, and this is also some kind of misunderstanding caused by the ambiguous descriptions in previous manuscript.

In fact, we do not calculate fidelity between classical shadows and MLE reconstruction in the whole algorithm. In SLST, the fidelity between proposed state τ (physical) and shadows $\hat{\rho}$ (unphysical) is used as loss function to optimize the parameters in τ . The accuracy of SLST is determined by the fidelity between returned state τ and target state. In single-photon experiment, the target state is pure state $\rho_{\gamma,\phi} = |\psi\rangle_{\gamma,\phi}\langle\psi|$. In two-photon experiment, the target state ρ_{η} is the MLE reconstruction with bulk optics (waveplates and polarizing beam splitters) and large amount of experimental runs ($M \approx 8 \times 10^5$). The reason we use MLE reconstruction with bulk optics as target state is that the bulk optical element can be considered as noise-free measurement device.

Fidelity $\hat{F}(\tau)$ is not an unbiased estimator with classical shadows $\{\hat{\rho}^{(m)}\}$ for mixed state τ , which is also pointed by other reviewers. In the revision, we use squared Frobenius norm $N_F(\rho, \tau) = \|\rho - \tau\|_F^2 = \text{Tr}(\rho - \tau)^2$ as loss function, where $\hat{N}_F(\tau)$ is an unbiased estimator with classical shadows $\{\hat{\rho}^{(m)}\}$. Figure 3 (c) and (d) are accordingly updated, where the fidelity between returned state τ and target state is presented. The results clearly indicate the superiority of SLST.

In the revision, we have rewritten this part to make a clear statement.

Comment 9: I was also not able to reproduce the MLE data. I simulated noisy MLE with $M=300$ and over 1000 instances the mean fidelity with the target state was 0.99 with a standard deviation of 1%. This is very much in line with the results for SLST and much better than what Fig 3 shows for MLE.

Reply: We appreciate the effort Reviewer made for checking our results. This is also some kind of misunderstanding caused by ambiguous data presentations in previous manuscript.

The metasurface we reported is also able to collect data (σ_x , σ_y and σ_z bases) for MLE reconstruction. However, the metasurface-induced measurement errors, which is pointed out by Reviewer, reduces the accuracy of MEL reconstruction with metasurface compared to that with bulk optical setting. In the comparison, SLST, SGQT and MLE reconstruct physical states τ from the data collected with metasurface (noisy) and the fidelity is calculated between τ and target state, where the target state (two-photon experiment) is obtained by MLE reconstruction with large amount of data ($M \approx 8 \times 10^5$) collected from bulk optical setting (almost perfect). We agree with Reviewer's simulation that MLE reconstruction with $M = 300$ runs achieves high fidelity with MLE reconstruction with large number of runs *under the same level of noise*.

The detection of state ρ with noisy measurement device can be effectively considered as a combination of noisy channel acting on ρ and ideal measurement device. Along this spirit, we simulate ideal SLST, SGQT and MLE reconstruction of noisy single-qubit state

$$\rho' = \mathcal{E}_D(\rho) = \frac{p}{2}\mathbb{1}_2 + (1-p)\rho, \quad (13)$$

where single-qubit pure state $\rho = |\psi_{\gamma,\phi}\rangle\langle\psi_{\gamma,\phi}|$ is depolarized with probability of p . The simulated results of fidelity between reconstruction and ρ is shown in Fig. 4. We observe that the fidelity with MLE recon-

Figure 4: **Simulated results of fidelity between ρ and reconstructions from SLST, SGQT and MLE with data collected from ρ' . (a), $p = 0.1$. (b), $p = 0.2$. (c), $p = 0.3$.**

struction converges quickly with a few hundreds of runs in the cases of $p = 0.1$, $p = 0.2$ and $p = 0.3$, while the accuracy (converged value) of reconstruction depends on p . In contrast, SLST and SGQT achieve higher accuracy as SPSA is robust to noise.

In revision, we have added a section in Supplementary Materials to clarify this point.

Comment 10: On page 8 the authors induce loss and discuss the resulting SNR. I am a bit puzzled by this since two-photon experiments are usually post-selected on coincidence detections, which should hardly be affected by the induced loss.

Reply: We thank Reviewer for this remark. We agree with Reviewer that two-photon experiments are usually post-selected on coincidence detections. As we mentioned in main text, the first-order SPDC emission is desired *signal* as it corresponds to the maximally entangled state (Bell state). However, the higher-order emission in SPDC, i.e., more than two photons are generated, also contributes to the coincidence in post-selection as the detector is not photon-number-resolved. The higher-order emission is thus regarded as *noises*, which reduces the fidelity between observed two-photon state with maximally entangled state. The photon loss significantly reduces the coincidences from first-order emission, but the reduction of coincidences from higher-order emission is relatively small. Consequently, the signal-to-noise ratio is reduced, which is reflected by the decrease of fidelity with (noisy) MLE reconstruction.

In the revision, we have clarified this point.

Comment 11: The manuscript talks about real-time shadow tomography, but I see no real-time data.

Reply: We thank Reviewer for this valuable remark, which is also raised by other reviewers. In revision, we have added the results of real-time estimation of expectations of observables in Figure 2(c), which is also shown below in Figure 5).

Figure 5: Real-time estimation of expectations of observables O , which are randomly selected from set \mathcal{O} .

Reply to the Report of Reviewer 3

Brief summary

In the manuscript, An et al. experimentally demonstrate using a single metasurface optical chip for implementing positive-operator-valued measures to perform classical shadow and full quantum state tomography. With the metasurface optical chip, they realize an octahedral POVM. First, they use this to study the sample complexity of classical shadow tomography estimating expectation values of observables in single photon states. Consistent with theoretical predictions, they find that their octahedral POVM outperforms the SIC POVM. Secondly, they demonstrate full tomographic reconstruction of single and two-photon states introducing a so-called self-learning shadow tomography. This is an iterative algorithm based on simultaneous stochastic perturbation and gradient descent. They find that this algorithm outperforms other techniques, and can be made more robust against experimental noise, using techniques from robust shadow tomography.

Manuscript content

The manuscript addresses a timely and interesting subject – improving efficiency of (classical shadow) tomography using non-trivial POVMs. The metasurface chip offers a novel way to implement such POVMs.

With respect to the content, I have the following questions and comments:

We appreciate the time and efforts Reviewer took to review our manuscript, and we thank Reviewer for her/his insightful comments and suggestions. In particular, the comment on characterization of metasurface-enabled POVM helps us to investigate the influence of metasurface-induced errors in different tomographic technologies. Besides, the comment that “fidelity is not a biased estimator with classical shadows” helps us to improve SLST, in which we set squared Frobenius norm (unbiased estimator) as loss function instead of fidelity. Below are the point-to-point responses to Reviewer’s comments.

Comment 1: Before studying the sample complexity of classical shadow tomography, it seems logical to first examine the capability to reconstruct desired expectation values accurately. Using the same data from Fig. 2a, it would be beneficial to clearly show that the expectation values of the observables in Fig. 2 are estimated reliably, rather than just focusing on their empirical variance.

Reply : We thank Reviewer for this valuable remark, and we agree with Reviewer that the accuracy of estimation should be examined first. In the revision, we have revised Figure 2. Specifically, the accuracy of shadow tomography with metasurface is shown in Figure 2b.

Comment 2: A direct characterization of the POVM is to some extent missing in the main text, or should be more explicitly commented on [as it is the paper’s main topic]. For instance, to which fidelity are the projector of 6 states measured? Showing the reconstructed expectation values in Fig. 2 would also serve to characterize the POVM more accurately. While the latter sections on tomography does touch upon this, it’s largely within the context of advanced projection and optimization techniques.

Reply: We thank Reviewer for this valuable remark, and we agree with Reviewer that the POVM should be characterized. In the revision, we have characterize the POVM from two aspects.

1. **The errors in reconstruction of Stokes parameters.** The metasurface we reported is a full-Stokes polarimetry device. Arbitrary polarization state can be represented by the Stokes vector formalism as $\mathbf{S} = [S_0, S_1, S_2, S_3]^T$. The elements S_i is defined by

$$\begin{aligned} S_0 &= I_H + I_V = I_+ + I_- = I_R + I_L, \\ S_1 &= I_H - I_V, \\ S_2 &= I_+ - I_-, \\ S_3 &= I_R - I_L, \end{aligned} \tag{14}$$

where I_l is the power intensity along polarization $l \in [H, V, +, -, R, L]$. We test our metasurface with six input polarizations $l \in [H, V, +, -, R, L]$, and record the distribution of power intensity on the focal plane for each input. According to the power intensity, we calculate the normalized Stokes parameter $\mathbf{s} = [s_1, s_2, s_3]$, where

$$\begin{aligned} s_1 &= \frac{I_H - I_V}{I_H + I_V}, \\ s_2 &= \frac{I_+ - I_-}{I_+ + I_-}, \\ s_3 &= \frac{I_R - I_L}{I_R + I_L}, \end{aligned} \tag{15}$$

and the results are shown in Table 2. We further calculate the measurement error defined between the reconstructed s_i and ideal s_i , which yields the average errors of s_1 , s_2 and s_3 are 0.097, 0.027 and 0.029 in simulation and 0.101 ± 0.005 , 0.086 ± 0.005 and 0.073 ± 0.005 in experiment.

Stokes parameters	s_1		s_2		s_3	
	Sim.	Exp.	Sim.	Exp.	Sim.	Exp.
$ H\rangle$	0.968	0.996 ± 0.001	0.025	-0.100 ± 0.016	0.024	-0.025 ± 0.017
$ V\rangle$	-0.991	-0.913 ± 0.003	-0.046	0.124 ± 0.016	-0.023	0.083 ± 0.016
$ +\rangle$	-0.120	-0.132 ± 0.016	0.988	0.936 ± 0.002	0.061	0.181 ± 0.016
$ -\rangle$	-0.140	0.148 ± 0.016	-0.986	-0.925 ± 0.002	-0.046	0.064 ± 0.016
$ R\rangle$	-0.147	0.163 ± 0.016	0.031	-0.122 ± 0.016	0.991	0.920 ± 0.003
$ L\rangle$	-0.133	-0.069 ± 0.016	-0.033	-0.034 ± 0.017	-0.992	-0.995 ± 0.001

Table 2: The simulation and experimental results of reconstructed Stokes parameters.

2. **The errors in estimation of expectations with classical shadows.** We use the classical shadows $\hat{\rho}$, which is obtained from metasurface-based POVM on a single $\rho_{\gamma,\phi}$, to estimate the expected values of 128 randomly selected O . The estimation error is defined by the distance between estimation and ideal value, i.e., $|\hat{O} - \langle O \rangle|$, and the maximal error is shown in Figure 2b. We observe that the error converges to 0.07 with increase of experimental runs, which is consistent with the measurement errors obtained from reconstruction of Stokes parameters.

The imperfection of metasurface indeed introduces errors in shadow tomography. However, this error can be eliminated—as reflected by experimental results—with SPSA optimization. SPSA formally accommodates noisy measurements of the objective function, which is an important practical concern in experiment.

In the revised manuscript, we have added the data to characterize metasurface-based POVM, along with explicit analysis.

Comment 3: It is unclear to me how the proposed SPSA optimization can be executed with data from randomized measurements if both, theory and experimental state, are mixed. Since in this case, the fidelity F is not a polynomial function of ρ , there does not exist an unbiased estimator for F using classical shadows. Is the presented algorithm thus specific to pure theory states [in which case the fidelity is a simple overlap] – if so this should be stated clearly – or can it be used for mixed theory states as well?

Reply: We thank Referee for this insightful remark, which indeed improves SLST. We agree with Referee that the fidelity $\hat{F}(\tau)$ is not a polynomial function of $\{\hat{\rho}^{(m)}\}$ for mixed state τ , which makes $\hat{F}(\tau)$ *not* an unbiased estimator with $\{\hat{\rho}^{(m)}\}$.

In the revision, we address this issue by using the squared Frobenius norm between τ and ρ

$$N_F(\tau, \rho) = \|\rho - \tau\|_F^2 = \text{Tr}(\rho - \tau)^2 = \text{Tr}(\rho^2) + \text{Tr}(\tau^2) - 2\text{Tr}(\rho\tau) \quad (16)$$

as a new cost function in SPSA. For this cost function, we indeed can write down the *unbiased* estimator with shadows $\{\hat{\rho}\}$ as follows.

For the first term, it shows

$$\frac{2}{M(M-1)} \sum_{m < n} \text{Tr} \left[\hat{\rho}^{(m)} \hat{\rho}^{(n)} \right], \quad (17)$$

which is an order-2 polynomial function of $\hat{\rho}$ [Nat. Phys. 16, 1050 (2020)]. Obviously, $\text{Tr}(\hat{\rho}\tau)$ is an *unbiased* estimator as well. Then, the *unbiased* estimator of N_F in total is

$$\hat{N}_F(\tau) = \frac{2}{M(M-1)} \sum_{m < n} \text{Tr} \left[\hat{\rho}^{(m)} \hat{\rho}^{(n)} \right] + \text{Tr}(\tau^2) - 2 \sum_m \text{Tr}(\hat{\rho}^{(m)}\tau). \quad (18)$$

Accordingly, the gradient formula in Eq.(4) is changed to

$$\mathbf{g}_k = \frac{\hat{N}_F(\mathbf{r}_k + \beta_k \mathbf{\Delta}_k) - \hat{N}_F(\mathbf{r}_k - \beta_k \mathbf{\Delta}_k)}{2\beta_k} \mathbf{\Delta}_k. \quad (19)$$

In the revision, we have revised the results of SLST using the squared Frobenius norm $\hat{N}_F(\tau)$ as cost function. As expected, SLST still exhibits advantages compared to MLE reconstruction, similar as the previous results based on fidelity.

Comment 4: The two photon states are not defined. Why are not both photons detected with a metasurface POVM? Connected to this, how is the scalability to detect many-photon states? Does one need a separate metasurface for each photon?

Reply: We thank Reviewer for this valuable remark.

The two photon states are not defined.

Ideally, the two-photon state is $|\psi\rangle_\eta = \sqrt{\eta}|HV\rangle + \sqrt{1-\eta}|VH\rangle$, where the parameter η is determined by the polarization of pump light, i.e., $\eta|H\rangle + \sqrt{1-\eta}|V\rangle$.

Why are not both photons detected with a metasurface POVM? Connected to this, how is the scalability to detect many-photon states? Does one need a separate metasurface for each photon?

The scope of our work is to validate the capability of metasurface to perform shadow tomography with POVM. As pointed out by Reviewer, the accuracy of estimation, as well as the sample complexity, with metasurface-enabled POVM is of interest in current investigation. We further investigate how to eliminate the metasurface-induced error in shadow tomography. The experimental demonstration with single photon—as reflected with experimental results—is sufficient for the first step along this line of research.

We agree with Reviewer that the detection of two-photon (spatially separated) with metasurface is of paramount importance for efficient characterization of scalable multiphoton states. Time multiplexing is a promising technology to generate scalable multiphoton states encoded in polarization degree of freedom [Phys. Rev. Lett. 129, 150501 (2022)], where *two* measurement devices are sufficient. However, the optical loss of metasurface should be taken into consideration when utilizing metasurface as an efficient measurement device along this technical roadmap. The overall optical loss of our metasurface is 48% with CMOS detection and 75% with fiber collection, respectively. The optical loss can be further reduced by high-order diffraction suppression [Nano. Lett. 17, 6267 (2017); Nat. Commun. 14, 5602 (2023)], multi-layer design [Light: Sci. Appl. 8, 78 (2019)] and symmetric division [Adv. Photonics. Res. 3, 2100373 (2022)] of metasurface. The relevant investigations are beyond the scope of our work.

In the revision, we have clarified this point in conclusion.

Comment 5: The robust shadow estimation should be explained in more detail, in particular since the two-photon states are detected with two different detectors. Also, robust shadow estimation makes certain assumptions about the noise. Are these fulfilled?

Reply: We thank Reviewer for this valuable comment. Indeed, in the two-photon robust shadow tomography, two photons are detected with different measurement devices, i.e., one is the metasurface-enabled POVM, while the other is in the traditional way, randomly detected on three Pauli basis (σ_x , σ_y and σ_z) which is realized by bulk optics. We would like to clarify that the mathematical models of these two (noisy) measurement devices are identical, so the framework of (multi-qubit) robust shadow tomography is still valid in such experimental setting, even though the specific noise type and strength of the two measurement devices may be different. In addition, a self-consistent introduction to the details of robust shadow tomography is given in our supplementary material section I B, where in particular we focus on the multi-qubit Pauli measurements as used in our experiment.

In robust shadow tomography proposal [PRX Quantum 2, 030348 (2021)], two main assumptions are made on the noise of measurement device to make the theoretical framework rigorous and analytical, i.e.,

A1. The noise in the circuit is gate-independent, time-stationary, Markovian noise.

A2. The experimental device can generate the computational basis state $|0\rangle \equiv |0\rangle^{\otimes N}$ with sufficient high fidelity.

Our experimental device is able to generate $|HH\rangle$ with high fidelity (> 0.99) so that **A2** is satisfied. The noise introduced by bulk optical elements are fixed (the accuracy of the angle of waveplates and splitting ratio of PBS) and generally quite small so that it can be considered as gate-independent, time-stationary and Markovian noise. On the other hand, Metasurface is a passive optical device so that the metasurface-induced noise is time-stationary and Markovian noise as well. Note that the measurement error by metasurface is not strictly gate-independent. As reflected by the results of reconstructed Stokes parameters, the errors in σ_x , σ_y , and σ_z measurements are 0.086 ± 0.005 , 0.073 ± 0.005 , and 0.101 ± 0.005 , respectively. The inhomogeneous noise retards the faithful reconstruction (fidelity=1) as proved in [PRX Quantum 2, 030348 (2021)], however, robust shadow tomography still works as shown in Figure 4 in main text.

In the revision, we have clarified this point.

Manuscript presentation

The manuscript is overall well-written and well-structured. However, details and proper definitions are sometimes missing or can only be guessed from context. This concerns for instance (but not only):

Comment 6: Throughout the manuscript, one should carefully distinguish between shadow tomography (Aaronson *et al.*) and classical shadow tomography (Huang *et al.*).

Reply: Indeed, Aaronson *et al.* coined the term-shadow tomography, however, the concrete implementation of Aaronson’s procedure requires exponentially long quantum circuits that act collectively on all the copies of the unknown state. Classical shadow tomography (Huang *et al.*) can be seen as a further combination of shadow tomography and traditional tomography based on least-square estimator. We appreciate this valuable point of clarification. In response to this comment, we would like to clarify that the “shadow tomography” referred in our manuscript specifically means “classical shadow tomography” (Huang *et al.*).

To avoid any confusion, we revise the manuscript. We provide a clear distinction between shadow tomography (Aaronson *et al.*) and classical shadow tomography (Huang *et al.*) in the second paragraph, and change our terminology to ensure that our terminology accurately reflects this choice throughout the updated manuscript.

Comment 7: Page 3, first paragraph: Should it read “Proportional to the projector to . . .”

Reply: We thank Reviewer for this remark and we have revised accordingly.

Comment 8: Page 3, first paragraph: It would be helpful to explain in more details why every 2-design gives rise to a POVM. A 1-design should be sufficient?

Reply: We thank the reviewer for this insightful comment about 2-design and POVMs. (Projective) t -design characterizes an ensemble of quantum states which can mimic the statistical property of Haar random states to t -th order function. Mathematically, that is,

$$\frac{1}{L} \sum_{l=1}^L |\psi_l\rangle \langle \psi_l|^{\otimes t} = \int |\psi\rangle \langle \psi| d\psi, \quad (20)$$

where there are L elements in the ensemble $\mathcal{E} = \{|\psi_l\rangle\}$. It is not hard to check that a t -design is also a $(t - 1)$ -design. For $t = 1$, the integral result is $\mathbb{1}/d$; for $t = 2$, the result is $(\mathbb{1}^{\otimes 2} + \mathbb{F})/d(d + 1)$, where

\mathbb{F} is the swap operator on 2-copy Hilbert space. From the ensemble \mathcal{E} , one can always build a POVM as $\mathbf{E} = \{E_l\}_{l=1}^L$ with the elements $E_l = \frac{d}{L} |\psi_l\rangle \langle \psi_l|$.

For the qubit-case considered in our work, a computational basis $\{|0\rangle, |1\rangle\}$ is 1-design, and the summation of the density matrix of them is proportional to the identity operator $\mathbb{1}$. Actually, any orthogonal basis $\{|\psi\rangle, |\psi_\perp\rangle\}$ is a 1-design. The measurement in such basis is actually a POVM and it is clear that they are *not* information-complete for tomography, as one only measures in one basis.

Both SIC POVM and octahedron POVM are 2-design, which enables POVM being information-complete for tomography. Equivalently, 2-design property enables one to construct least-square estimator of the unknown state ρ by

$$\hat{\rho}_l = \mathcal{M}^{-1}(|\psi_l\rangle \langle \psi_l|) = 3|\psi_l\rangle \langle \psi_l| - \mathbb{1}_2. \quad (21)$$

The proof of this result is elaborated in our Supplementary Material I A.

Comment 9: Page 4, last paragraph: **O** and **P** are not defined.

Reply: We thank Reviewer for this remark. **O** is set of 128 observables in form of single-qubit projectors, while **P** is the set of 20 experimentally prepared $|\psi_{\gamma,\phi}\rangle$.

In the revision, we have clarified the definition of **O** and **P**.

Comment 10: Page 5: How is the maximum over **P** computed?

Reply: We thank Reviewer for this remark. Experimentally, it is impossible to maximize $\text{Var}(\hat{\delta})$ over all possible ρ . We experimentally prepare 20 $|\psi_{\gamma,\phi}\rangle$ that are uniformly distributed on the Bloch sphere, forming the set **P**. For each prepared state, we perform shadow tomography. Then, the maximization of $\text{Var}(\hat{\delta})$ is taken over the set **P**. In the revision, we have clarified the maximization accordingly.

Comment 11: Page 5: SIC POVM is not defined.

Reply: We thank Reviewer for this remark. In the revision, the explicit form of SIC POVM is presented in Supplementary Materials.

REVIEWER COMMENTS

Reviewer #1 (Remarks to the Author):

Authors make a good job in the revision. I recommend its publication on NC.

Reviewer #2 (Remarks to the Author):

The authors have made a very good effort to respond to all the concerns raised by myself and the other reviewers. There are a few minor questions that remain and that should be answered before publication.

1) Noisy measurements.

With a readout error on the order of 10%, the measurements are very noisy, and significantly reducing the noise seems quite challenging. I understand that under certain assumptions this noise can be inverted in post-processing. However, any such procedure becomes highly sensitive to the noise calibration and great care has to be taken to enable faithful estimation in such a situation.

It should be stated more clearly in the main text, how noise is taken into account in the reconstruction.

From Appendix I.B. it appears that the way this is done is by using a device calibration from which a noise channel is extracted that is inverted to construct the classical shadow. There are a few points of concern here.

First, the calibration accuracy should enter into the uncertainties of the final estimates.

Second, I don't quite understand why the same procedure cannot be used for improving the accuracy of the estimation of observables and the MLE results.

Third, it appears that the noise of the device does not satisfy the assumptions of the above procedure. A critical discussion of this and maybe some classical simulations is needed. Related to this, it appears from Fig.4 in the Appendix that the imperfections are strongly biased in the HV basis, which is typically more problematic than depolarizing noise. To what extent does this cause issues?

2) Cost Function.

The choice of the Frobenius norm as cost function is interesting.

Understandably, fidelity is not ideal for comparing against mixed states, however typically one is interested in pure states as the ideal state. In this case, I would suspect that the fidelity is computationally much more efficient than the Frobenius norm and also converges much faster since it is a linear function. A comment about the computational cost of the optimization would be useful.

3) Accuracy.

On page 8 it is claimed that Fig.3 shows that for small M , MLE is more accurate than SGQT.

Since these are experimental results, measuring accuracy by the fidelity with the target state is not appropriate. Later on, it seems that fidelity is actually measured with the reconstructed state from bulk-optics MLE with many shots. If this is the case it should be stated clearly.

Alternatively, theory simulations with a known ground truth state would help as in the new appendix V.C.

4) Other.

It is repeatedly stated that "SPSA is especially efficient in high-dimensional problems in terms of providing a good solution for a relatively small number of measurements of the objective function". I do not understand this. What is high-dimensional referring to here?

Summary of Changes

In response to Reviewer #2's comments and suggestions, we have made the following changes.

1. On Page 7, we have changed “high-dimensional” to “multi-parameter” to avoid misunderstanding.
2. On Page 7, we have added two sentences to discuss the computational cost of SPSSA and MLE.
3. On page 8, we have modified the sentence to clarify the reason we select ideal state as target state in fidelity calculations.
4. On page 9, the first paragraph, we have clarified the SPSSA can accommodate noisy measurements of the cost function.
5. On page 9, the second paragraph, we have modified several sentences to clarify the reason we select MLE reconstruction as target state in fidelity calculations.
6. On page 10, we have updated Figure 4, where the statistic errors in calibration is considered.
7. On page 10, we have revised the first paragraph to clarify: 1) why the robust shadow tomography (calibration) is valid in our experiment and 2) why the enhancement of robust SLST is significant at high-level optical loss.
8. We have added a section (I.C) in Supplementary Materials, where the simulations of SLST and robust SLST with gate-dependent measurement noise are provided.

All the changes have been highlighted in blue. Below, please find our point-to-point responses (in black) to Reviewer #2's report (in blue).

Reply to the Report of Reviewer #2

The authors have made a very good effort to respond to all the concerns raised by myself and the other reviewers. There are a few minor questions that remain and that should be answered before publication.

We appreciate the time and effort Reviewer #2 took to review our manuscript. In particular, her/his comments on the calibration of SLST help us to deepen the understanding of robust SLST. We are grateful to Reviewer #2 for her/his recommendation for publication after minor revisions.

Comment 1: 1) Noisy measurements.

With a readout error on the order of 10%, the measurements are very noisy, and significantly reducing the noise seems quite challenging. I understand that under certain assumptions this noise can be inverted in post-processing. However, any such procedure becomes highly sensitive to the noise calibration and great care has to be taken to enable faithful estimation in such a situation.

It should be stated more clearly in the main text, how noise is taken into account in the reconstruction. From Appendix I.B. it appears that the way this is done is by using a device calibration from which a noise channel is extracted that is inverted to construct the classical shadow.

Reply 1: We thank Reviewer for this remark, and we agree with Reviewer that “It should be stated more clearly in the main text, how noise is taken into account in the reconstruction”. The metasurface-induced measurement errors can be suppressed by SLST as SPSA can accommodate noisy measurements of the cost function, which is reflected by the results shown in Figure 2 and Figure 3. By introducing the optical loss, the fidelity of prepared state is reduced, which is equivalent to increasing the measurement errors. Then, the robust SLST (calibration procedure) is capable to suppress the higher level noise as reflected in experimental results in Figure 4 and the simulations Figure 1 in Reply 4.

There are a few points of concern here.

Comment 2: First, the calibration accuracy should enter into the uncertainties of the final estimates.

Reply 2: We thank Reviewer for this valuable remark. Indeed, the statistic error in calibration should be taken into account in the final estimation. In revision, we have updated the statistic error of robust SLST in Figure 4.

Comment 3: Second, I don't quite understand why the same procedure cannot be used for improving the accuracy of the estimation of observables and the MLE results.

Reply 3: We thank Reviewer for this remark. First, we would like to clarify the aims of original shadow tomography [Huang *et al.* Nature Physics, 16, 1050 (2020)], SLST and MLE. SLST and MLE aim to reconstruct the unknown quantum state by solving optimization problems, while shadow tomography directly estimates (linear) observables without requiring the full information of unknown states. Accordingly, the techniques of these schemes are different. The enhancement of accuracy in SLST attributes to SPSA algorithm on the corresponding collected data. In fact, shadow tomography is the optimal scheme to estimate observables in ideal (noiseless) case. When the measurements in shadow tomography is noisy, robust shadow tomography [Chen *et al.* PRX Quantum 2, 030348 (2021)] is able to suppress the noise-induced inaccuracy in estimation of observables.

Secondly, we would like to clarify the difference between SLST and MLE. The MLE reconstruction aims to reconstruct the unknown states by minimizing the following maximal-likelihood cost function using a gradient-based optimization method

$$\begin{aligned} \text{minimize} \quad & \mathcal{L}(\tau) = \left\| W \left(\sum_j (\text{Tr}(|j\rangle \langle j| \tau) - f_j) |j\rangle \right) \right\|^2 \\ \text{subject to} \quad & \tau \geq 0, \text{Tr}(\tau) = 1, \end{aligned} \quad (1)$$

where $|j\rangle$ is the measurement basis, f_j is the empirical frequency on $|j\rangle$, and W is a diagonal matrix of statistical weights W . The computational expense required to estimate gradient direction is directly proportional to the number of unknown parameters ($4^N - 1$ for an N -qubit state) as it approximates the gradient by varying one parameter at a time, which becomes an issue when the number of qubit is large. In SLST, the reconstruction is achieved by minimizing the cost function (Frobenius norm) using SPSA optimization, in which all parameters are simultaneously perturbed at one time and one gradient evaluation requires only two evaluations of the cost function. Thus, SLST is efficient compared to MLE in term of computational cost.

Comment 4: Third, it appears that the noise of the device does not satisfy the assumptions of the above procedure. A critical discussion of this and maybe some classical simulations is needed.

Reply 4: We thank Reviewer for this valuable remark. In robust shadow tomography [PRX Quantum 2, 030348 (2021)], two main assumptions are made on the noise of measurement device to make the theoretical framework rigorous and analytical, i.e.,

A1. The noise in the circuit is gate-independent, time-stationary, Markovian noise.

A2. The experimental device can generate the computational basis state $|0\rangle \equiv |0\rangle^{\otimes N}$ with sufficiently high fidelity.

A2 is satisfied since our experimental device is capable to generate $|00\rangle$ state ($|HH\rangle$) with high fidelity (> 0.99). The measurement errors induced by metasurface are not strictly gate-independent, as the errors in σ_x , σ_y , and σ_z measurements are 0.086 ± 0.005 , 0.073 ± 0.005 , and 0.101 ± 0.005 , respectively, so that **A1** is not strictly satisfied. Note that the strongest assumption **A1** in theory helps to obtain rigorous and analytical results. However, the theoretical framework still works for gate-dependent noise as shown by numerical simulation in Figure. 7 in Ref. [PRX Quantum 2, 030348 (2021)]. Our experimental results (Figure 4 in main text) confirmed this claim. To further confirm this point, we simulate the SLST and robust SLST on single-qubit state

$$\rho' = \left(1 - \frac{\delta_x + \delta_y + \delta_z}{3} \right) \rho + \frac{\delta_x}{3} \sigma_x \rho \sigma_x + \frac{\delta_y}{3} \sigma_y \rho \sigma_y + \frac{\delta_z}{3} \sigma_z \rho \sigma_z. \quad (2)$$

In each single run in simulation, the value of $\delta_{x,y,z}$ is randomly resampled from Gaussian distribution with mean value of $\bar{\delta}$ and standard deviation of σ . The simulation is equivalent to performing measurement with gate-dependent noises on the ideal state ρ . As shown in Figure 1 (a), the enhancement of robust SLST is not obvious when the noise is weak ($\bar{\delta} = 0.05$). Robust SLST exhibits advantage when the noise is strong ($\bar{\delta} = 0.1$) as shown in Figure 1 (b). The simulation results agree well with the results shown in Figure 4 in main text, where the robust SLST significantly enhances the accuracy when optical loss is high. In revision, we have added the simulation in Supplementary Materials.

Comment 5: Related to this, it appears from Fig.4 in the Appendix that the imperfections are strongly biased in the HV basis, which is typically more problematic than depolarizing noise. To what extent does this cause issues?

Figure 1: The simulation of robust SLST and SLST on the state in Eq. 2. (a), The simulation is performed by setting $\bar{\delta} = 0.05$ and $\sigma \in [0, 0.05]$ with interval of 0.01. (b), The simulation is performed by setting $\bar{\delta} = 0.1$ and $\sigma \in [0, 0.1]$ with interval of 0.02. The simulation is carried out with $M = 2000$ runs. In robust SLST, additional $M' = 2000$ runs are used for calibration. We set $k = 100$, hyperparameters $a_1 = 34.1$ and $b_1 = 5.7$.

Reply 5: We thank Reviewer for this remark. To simulate the measurements with metasurface-induced biased errors, we set $\delta_x = 0.05, \delta_y = 0.17$ and $\delta_z = 0.02$ in Eq. 2. The measurements of σ_x, σ_y and σ_z on this noisy state give the outcomes similar to noisy measurements on ideal state. The simulation results of robust SLST and SLST on the noisy state are shown in Figure 2. We observe that robust SLST achieves a better reconstruction when increasing M , which confirms the feasibility of robust SLST with noisy measurement.

Figure 2: The results of robust SLST and SLST on states in Eq. 2 with $\delta_x = 0.05, \delta_y = 0.17$ and $\delta_z = 0.02$, respectively. The simulation is carried out with $M = 2000$ runs. In robust SLST, additional $M' = 2000$ runs are used for calibration. We set $k = 100$, hyperparameters $a_1 = 34.1$ and $b_1 = 5.7$.

2) Cost Function.

The choice of the Frobenius norm as cost function is interesting.

Comment 6: Understandably, fidelity is not ideal for comparing against mixed states, however typically one is interested in pure states as the ideal state. In this case, I would suspect that the fidelity is computationally much more efficient than the Frobenius norm and also converges much faster since it is a linear function. A comment about the computational cost of the optimization would be useful.

Reply 6: We thank Reviewer for this remark. For the target state τ being pure state, the squared Frobenius norm $N_F(\tau, \rho) = \|\rho - \tau\|_F^2$ is related to fidelity $F(\tau, \rho) = \text{Tr}(\tau\rho)$ by

$$\begin{aligned}
 N_F(\tau, \rho) &= \|\rho - \tau\|_F^2 \\
 &= \text{Tr}(\rho\tau)^2 \\
 &= \text{Tr}(\rho^2) + \text{Tr}(\tau^2) - 2\text{Tr}(\rho\tau) \\
 &= \text{Tr}(\rho^2) + 1 - 2F(\tau, \rho).
 \end{aligned} \tag{3}$$

If we use fidelity $F(\tau, \rho)$ as cost function, there is no difference when calculating the gradient \mathbf{g}_k except a factor of -2 which can be absorbed in β_k . Thus, the convergency of SLST should be same. To confirm this, we perform SLST with cost function of squared Frobenius norm and fidelity respectively, on the two-photon data to reconstruct pure state τ_k . As shown in Figure 3, the convergency of SLST with cost function of squared Frobenius norm and fidelity are the same.

Figure 3: The results of SLST to reconstruct two-photon state with cost function of N_F (red dashed lines) and F (blue dashed lines), respectively. The proposed state τ_k is set to be pure state.

3) Accuracy.

Comment 7: On page 8 it is claimed that Fig.3 shows that for small M, MLE is more accurate than SGQT. Since these are experimental results, measuring accuracy by the fidelity with the target state is not appropriate. Later on, it seems that fidelity is actually measured with the reconstructed state from bulk-optics MLE with many shots. If this is the case it should be stated clearly. Alternatively, theory simulations with a known ground truth state would help as in the new appendix V.C.

Reply 7: We thank Reviewer for this remark. The accuracy of reconstruction with different techniques (SLST, SGQT and MLE) is characterized by the fidelity between reconstruction and target state. The target state should be very close to the experimentally prepared state. Along this spirit, the fidelity between prepared single-photon state and ideal state (pure state) $|\psi_{\gamma,\phi}\rangle$ is quite high ($> 99\%$), so that it is reasonable to set $|\psi_{\gamma,\phi}\rangle$ as target state in fidelity calculations. In the two-photon experiments, the prepared two-photon state is affected with much more noises including the high-order emission in SPDC and mode mismatch in Sagnac interferometer. The fidelity between prepared state and ideal state (pure state) $|\psi\rangle_\eta =$

$\sqrt{\eta}|HV\rangle + \sqrt{1-\eta}|VH\rangle$ is about 93% so that it is inappropriate to set $|\psi\rangle_\eta$ in fidelity calculations. We reconstruct the prepared two-photon state ρ_η using MLE with a large amount of collected data from bulk-optics setting (almost noiseless), and set the reconstructed state ρ_η as target state in fidelity calculation. In revision, we have made a clear statement.

4) Other.

Comment 8: It is repeatedly stated that “SPSA is especially efficient in high-dimensional problems in terms of providing a good solution for a relatively small number of measurements of the objective function”. I do not understand this. What is high-dimensional referring to here?

Reply 8: We thank Reviewer for the remark. In optimization, the “dimension” refers to the number of parameters evolved in the computation of gradient. The SPSA is a class of optimization algorithms which compute the gradient by perturbing all the parameters simultaneously. Therefore, SPSA is particularly well suited to problems involving large number of parameters. The state reconstruction of N -qubit state is such a problem, in which the optimization is taken place over $4^N - 1$ parameters. In revision, we have replaced “high-dimensional” with “multi-parameter” to avoid misunderstanding.

REVIEWER COMMENTS

Reviewer #2 (Remarks to the Author):

In the latest revision, the authors have resolved a range of concerns. A few points remain, see below, but otherwise, I believe the manuscript is suitable for publication.

1) Re calibration uncertainty: The updated caption of Fig.4 states that statistical noise was taken into account, not calibration uncertainties.

2) "Thus, SLST is efficient compared to MLE in term of computational cost."--- That also depends on the number of parameter variations that are needed. It could, in principle, be that although SLST varies fewer parameters per evaluation, it takes much longer to converge

3) Noise robustness: Firstly, the manuscript still does not explain how the protocol is robust against measurement noise. I believe, this is achieved like robust shadow estimation by inverting the noise channel characterized on the 0-state. However this must be explained explicitly in the main text. Secondly, I remain a bit skeptical about this procedure, since it appears to easily overestimate the quality of the state. In particular, it is very difficult to separate state preparation from measurement errors. e.g. if the photon source produces low-fidelity states due to some dephasing effect, the noise calibration would attribute this dephasing to the measurement and still return a state-fidelity close to 1. The data seems to reflect this behaviour since the estimated fidelity tends to increase with the number of shots, whereas the standard linear inversion (unbiased) estimator tends to fluctuate around the true value with increasing shot number.

Reviewer #2 (Remarks on code availability):

The available data seems to cover what is in the paper. However, github is not ideal as a repository. I suggest uploading the data to a permanent repository that assigns a DOI, such as Zenodo.

Reviewer #3 (Remarks to the Author):

The authors made substantial revisions and addressed my comments satisfactorily. Once the comments of the other referees have been addressed, I recommend publication.

Summary of Changes

In response to Reviewer #2's comments and suggestions, we have made the following changes.

1. On page 7, we have clarified the efficiency of SPSA.
2. On page 8, we have clarified the limitation of SPSA.
3. On page 10, we have explicitly explained the calibration process and its validation in our experiment.
4. On page 12, we have clarified the availability of data and code related to the study on Zenodo.

All the changes have been highlighted in blue. Below, please find our point-to-point responses (in black) to Reviewer #2's report (in blue).

Reply to the Report of Reviewer #2

Comment 1: 1) Re calibration uncertainty: The updated caption of Fig.4 states that statistical noise was taken into account, not calibration uncertainties.

Reply 1: We thank Reviewer for this remark. There are two kinds of “uncertainties” associated with calibration, i.e., systematic errors and statistical errors, respectively. The systematic errors in calibration are related to the input state $|HH\rangle$. In our experiment, the calibration is performed with ρ_{HH} reconstructed from MLE. Indeed, the MLE itself introduces systematic errors [Phys. Rev. Lett. 114, 080403 (2015)]. However, such bias is unavoidable when a *physical state* is necessary in calibration. It is an open question whether the calibration can be achieved with linear evaluations (similar concern is also arised in Comment 3), so that we cannot directly observe the systematic error in experiment.

We agree with Reviewer that systematic errors are important in calibration, and we investigate this problem in a numerical approach. The detection of state $\rho = |+\rangle\langle +|$ with noisy measurement device can be effectively considered as performing ideal measurements on state ρ passing through a depolarizing channel

$$\rho' = \mathcal{E}_D(\rho) = \frac{p}{2}\mathbb{1}_2 + (1-p)\rho. \quad (1)$$

We first calibrate the noisy measurement $\widetilde{\mathcal{M}}$ with ideal $\rho = |0\rangle\langle 0|$ and ρ_0^{MLE} respectively, where ρ_0^{MLE} is the reconstruction from MLE. Then we perform SLST with the calibrated $\widetilde{\mathcal{M}}$, and the results are shown in Figure 1. The MLE slightly underestimates fidelity, which is in agreement with the results in [Phys. Rev. Lett. 114, 080403 (2015)].

In revision, we have clarified this point.

Figure 1: Histogram of the fidelity with robust SLST among 500 repeats for different calibrations. For each repeat, $M = 500$ measurements are used for state reconstruction, and $M' = 500$ measurements are used for calibration.

Comment 2: 2) “Thus, SLST is efficient compared to MLE in term of computational cost.”— That also depends on the number of parameter variations that are needed. It could, in principle, be that although SLST varies fewer parameters per evaluation, it takes much longer to converge.

Reply 2: We thank Reviewer for this remark. We agree with Reviewer SLST takes much longer to converge compared to MLE. In Figure 2, we show the convergence of SLST and MLE in two-photon experiment. Although SLST costs much iterations to converge, SLST returns states with higher fidelity. Similar result

has been reported in SGQT [Phys. Rev. Lett. 117, 040402 (2016)]. Either SLST or SGQT employs SPSA for estimation. In optimization, one can terminate the algorithm if the returned result is converged or the assigned number of iterations has been reached. In the second case, SLST is efficient in term of computational cost as it returns a state with higher fidelity in small number of iterations.

In revision, we have clarified this point.

Figure 2: The results of experimental reconstruction of two-photon state with SLST and MLE. For each repeat, $M = 500$ measurements are used for state reconstruction, and $M' = 500$ measurements are used for calibration.

3) Noise robustness:

Comment 3: Firstly, the manuscript still does not explain how the protocol is robust against measurement noise. I believe, this is achieved like robust shadow estimation by inverting the noise channel characterized on the 0-state. However this must be explained explicitly in the main text.

Reply: We thank Reviewer for this remark. Referee is correct. The calibration is exactly the robust shadow tomography. In robust SLST, we first calibrate the noisy measurement apparatus (attenuator and metasurface) with high-fidelity $|HH\rangle$, and obtain the noisy channel $\tilde{\mathcal{M}}$. Then we perform SLST with collected data and noisy channel $\tilde{\mathcal{M}}$.

In revision, we have explicitly explained this part in the main text.

Secondly, I remain a bit skeptical about this procedure, since it appears to easily overestimate the quality of the state. In particular, it is very difficult to separate state preparation from measurement errors. e.g. if the photon source produces low-fidelity states due to some dephasing effect, the noise calibration would attribute this dephasing to the measurement and still return a state-fidelity close to 1. The data seems to reflect this behaviour since the estimated fidelity tends to increase with the number of shots, whereas the standard linear inversion (unbiased) estimator tends to fluctuate around the true value with increasing shot number.

Reply: We thank Reviewer for this valuable remark. One critical assumption in robust shadow tomography is that the experimental setup is able to generate $|0\rangle^{\otimes N}$ with sufficiently high fidelity. If the setup generates low-fidelity states, the calibration indeed overestimates the fidelity of the returned state. This is confirmed by the following simulations. An experimentalist would like to generate target state ρ with the setup, however, the imperfect setup generates noisy state $0.8\rho + 0.2\mathbb{1}_2/2$ instead of ρ . Meanwhile, the measurement is noisy as well. The experimentalist first calibrates the noisy measurement in Eq. 1 with generated $0.8|0\rangle\langle 0| + 0.2\mathbb{1}_2/2$, and obtains the biased $\tilde{\mathcal{M}}$. If one use biased $\tilde{\mathcal{M}}$ to reconstruct state, say $0.8|+\rangle\langle +| + 0.2\mathbb{1}_2/2$, SLST

overestimates the fidelity as shown in Figure 3, where the fidelity is calculated between returned state and $|+\rangle$.

In our experiment, the input state $|HH\rangle$ for calibration is with high fidelity (0.9956 ± 0.0005) so that this bias is significantly reduced. We agree with Reviewer that optimization algorithms, such as MLE and SPSA, would inevitably introduce bias in calculation of functions (such as fidelity) with reconstructed state [Phys. Rev. Lett. 114, 080403 (2015)]. In fact, if one only cares about the fidelity instead of the reconstruction of a *physical* state, the shadow tomography with metasurface-enabled POVM is an efficient (in term of sample complexity as we shown in Figure 2 in main text) and unbiased estimator as the fidelity with respect to target state can be decomposed into observables.

Also, we agree with Reviewer that the linear inversion is an unbiased estimator of quantum properties of unknown state. However, whether the calibration can be achieved with linear inversion is still an open question.

In revision, we have clarified the limitation of SLST.

Figure 3: The fidelity between returned state from robust SLST and $|+\rangle\langle+|$ (red dots). The theoretical fidelity between $0.8|+\rangle\langle+| + 0.2\mathbb{1}_2/2$ and $|+\rangle$ is represented with blue dashed line. For each run, $M = 500$ measurements are used for state reconstruction, and $M' = 500$ measurements are used for calibration. p represents the depolarizing rate in noisy channel \mathcal{M} .

Comment 4: (Remarks on code availability):

The available data seems to cover what is in the paper. However, github is not ideal as a repository. I suggest uploading the data to a permanent repository that assigns a DOI, such as Zenodo.

Reply 4: We thank Reviewer for this suggestion, we have uploaded the available data and code on Zenodo. <https://zenodo.org/records/10674374>

REVIEWERS' COMMENTS

Reviewer #2 (Remarks to the Author):

In the latest revision, the authors have carefully addressed the remaining concerns and made appropriate remarks in the manuscript to make the readers aware of these limitations. I believe the manuscript is now suitable for publication in Nature Communications.